# Adjusting agricultural emissions for trade matters for climate change mitigation

Adrian Foong[1,2,3], Prajal Pradhan [1✉], Oliver Frör [2] & Jürgen P. Kropp [1,4,5]

Reducing greenhouse gas emissions in food systems is becoming more challenging as food is increasingly consumed away from producer regions, highlighting the need to consider emissions embodied in trade in agricultural emissions accounting. To address this, our study explores recent trends in trade-adjusted agricultural emissions of food items at the global, regional, and national levels. We find that emissions are largely dependent on a country's consumption patterns and their agricultural emission intensities relative to their trading partners'. The absolute differences between the production-based and trade-adjusted emissions accounting approaches are especially apparent for major agricultural exporters and importers and where large shares of emission-intensive items such as ruminant meat, milk products and rice are involved. In relative terms, some low-income and emerging and developing economies with consumption of high emission intensity food products show large differences between approaches. Similar trends are also found under various specifications that account for trade and re-exports differently. These findings could serve as an important element towards constructing national emissions reduction targets that consider trading partners, leading to more effective emissions reductions overall.

[1] Potsdam Institute for Climate Impact Research (PIK), Member of the Leibniz Association, P.O. Box 60 12 03, D-14412 Potsdam, Germany. [2] University of Koblenz-Landau, Institute for Environmental Sciences, Landau, Germany. [3] adelphi research gemeinnützige GmbH, Berlin, Germany. [4] University of Potsdam, Institute for Environmental Science and Geography, Potsdam, Germany. [5] Bauhaus Earth gGmbH, Berlin, Germany. ✉email: pradhan@pik-potsdam.de

Food systems are a major driver of climate change, emitting 21–37% of global anthropogenic greenhouse gases (GHGs), or 10.8–19.1 Gt $CO_{2e}$/yr, during the period of 2007–2016[1,2]. Thus, to limit global warming well below 2 °C as stipulated in the Paris Agreement, emissions need to be drastically reduced at every stage of the food system from pre-production to post-consumption[3]. While the share of emissions contributed by each stage depends on food items, the largest share generally stems from the agricultural production or farm-gate stage[4–6].

A key element in international agreements over climate policies and emissions reduction is how countries report their respective national emissions[7]. To date, these accounting approaches only consider emissions produced within a country's borders (i.e., production-based accounting), a legacy of the Kyoto Protocol and emissions accounting frameworks provided by the Intergovernmental Panel on Climate Change (IPCC)[8–10]. However, with food increasingly being traded internationally and consumed away from where production takes place, the role of trade in national emissions accounting cannot be ignored. Equally important is the need to understand how the different specifications of including trade in emissions accounting may vary, considering the differential roles of producer, consumer, and intermediary trading countries in reducing emissions. Indeed, previous studies have discussed and estimated the disparity in results between the conventional production-based emissions accounting approach versus a more consumption-based approach in which trade (i.e., imports and exports) is also adjusted for[7,9,11–14].

A limited number of studies have applied trade-adjusted approaches to estimate emissions from the agricultural sector[15–19]. Most emissions accounting approaches look at emissions either from a producer or consumer perspective with the role of intermediary trading countries being largely overlooked. A broad overview of food-related agricultural emissions that accounts for trade flows at national levels remains a major knowledge gap in GHG emission inventories[4].

Thus, we aim to explore, for all food items at the global, regional, and national levels (Supplementary Tables 1, 2), recent trends from 1986 to 2017 in (i) trade-adjusted agricultural emissions and how they differ from the production-based approach, and (ii) emissions embodied in trade flows between producer and consumer regions. Our analyses focus on agricultural emissions from the production stage (i.e., within the farm-gate). We used data from the Food and Agriculture Organization Statistical Database (FAOSTAT), which provides comprehensive open access data on various aspects of food and agriculture at the country level. We then adjusted these agricultural production emissions using FAOSTAT's emission intensities and trade data (in metric tonnes), and applied a bilateral trade input-output (BTIO) approach (see Methods). We define trade-adjusted agricultural emissions as the sum of production-based emissions and import emissions minus export emissions[13]. Additionally, considering the range of alternative specifications that account for emission intensities and trade differently, we conducted three sensitivity analyses. The first considered how replacing regional emission intensities by global ones for non-producer countries would change our estimations of emissions. The second adopted a technology-adjusted approach, as suggested by Kander et al.[20]. The third considered the emission intensities associated with both production and imports to account for re-exports. Separately, we also tested how agricultural land-use emissions, using FAOSTAT data on emissions from forestry and other land use (FOLU), would change when adopting a trade-adjustment approach. In all analyses, we show values in terms of three-year averages, unless specified otherwise.

## Results

**Trade-adjusted agricultural emissions**. Global trade-adjusted agricultural emissions (TAE), in total absolute terms, increased from 3.86 Gt $CO_{2e}$/yr in 1987 to 5.02 Gt $CO_{2e}$/yr in 2015. This increase in emissions corresponds to the rise in the global agricultural production volumes driven by population growth and changing diets, resulting in an increase in production-based emissions (PBE) (Supplementary Figs. 1, 2). However, in per capita terms, we found a decrease in TAE from 0.77 t $CO_{2e}$/cap/yr in 1987 to 0.68 t $CO_{2e}$/cap/yr in 2015. This fall in per capita TAE is because of the increase in agricultural production that is largely driven by efficiency improvements afforded by agricultural intensification[21,22], thus resulting in a decrease in emission intensity (Supplementary Fig. 3).

Similarly, total absolute TAE also increased in most regions, with the exceptions of Europe, Oceania and the Former Soviet Union where TAE declined over time (Supplementary Fig. 4). Such exceptions reflect the increase in agricultural productivity and high resource-use efficiency in Europe, as well as the low input systems that characterise meat production in Oceania[21]. However, the drop in total absolute TAE in the Former Soviet Union is the consequence of the collapse of the Soviet Union in 1991, and the subsequent economic downturn and removal of subsidies for beef production and consumption[19,23].

At the country level, some of the highest total absolute TAEs were found in several countries in Asia, North America, and Latin America and the Caribbean (Fig. 1). For example, in mainland China, total absolute TAE rose from 452.2 Mt $CO_{2e}$/yr in 1987 to 705.2 Mt $CO_{2e}$/y in 2015 – the largest TAE among all countries in 2015. Moreover, a number of countries in Africa and other parts of Asia also saw some of the largest increases in total absolute TAE between 1987 and 2015. In Pakistan, this jump was 60.0 Mt $CO_{2e}$/yr, whereas in Nigeria, the increase was 41.6 Mt $CO_{2e}$/yr between 1987 and 2015. These trends reflect the economic growth, rising populations, and dietary shifts toward meat-rich diets seen in many of these regions[24–26].

In terms of per capita TAE, the distribution was somewhat different from the total absolute TAEs (Fig. 1). Countries in Oceania and Latin America and the Caribbean had some of the highest per capita TAE (e.g., Australia's and Uruguay's per capita TAEs were 2.41 and 3.77 t $CO_{2e}$/cap/yr respectively in 2015), reflecting these countries' higher affluence and consumption of animal-source foods, particularly of ruminant meat.

Additionally, a number of countries scattered in other regions, particularly those that fall under the lists of emerging and developing economies or low-income developing countries[27], also displayed high per capita TAE (>3.0 t $CO_{2e}$/cap/yr in 2015), most notably Mongolia and countries in southern and central Africa. These results show that per capita TAE is not always directly related to economic development and affluence as it is sometimes implied for consumption in general[9], and may simply be an indication of high consumption of more emission-intensive food items or large agricultural emission intensities. It is also notable that most countries in Asia, despite having relatively large total absolute TAEs, generally had low levels of per capita TAE (<0.5 t $CO_{2e}$/cap/yr in 2015 on average) due to their larger populations and relatively low food consumption per capita, particularly for countries such as mainland China, India, and Japan.

When comparing PBE with TAE, we found that the differences were large for certain countries (Fig. 2; see also Supplementary Fig. 5 for per capita results). These large differences were especially the case for several major importing countries where TAE was much greater than PBE. Two such cases include city-states such as Hong Kong and Singapore, where TAEs were more than 50 times larger than their respective PBEs in 2015. This disparity also holds for other import-dependent economies such as those in the Middle

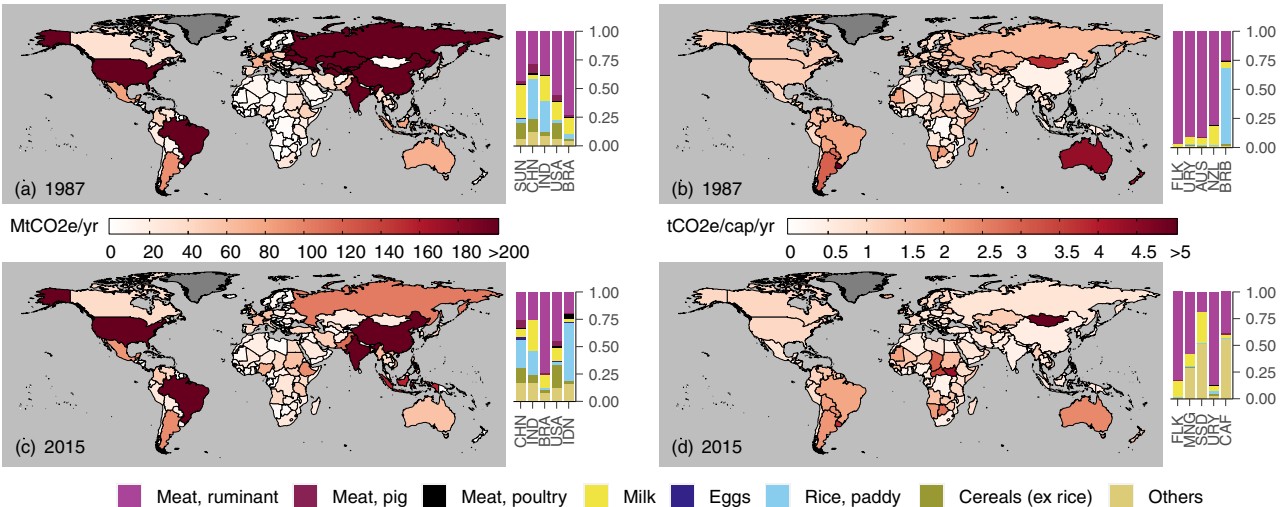

**Fig. 1 Trade-adjusted agricultural emissions (TAEs) in 1987 and 2015.** Total absolute TAEs are shown on the left (**a**, **c**), and per capita TAEs are shown on the right (**b**, **d**). A darker red colour indicates higher TAE levels, whereas dark grey colours indicate countries with no available data (e.g., Greenland). In addition, the bar graphs show the top five countries with the largest total absolute and per capita TAEs, in decreasing order from left to right. The bars also consist of the relative contributions of each food group to these TAEs. Country names are shown according to their respective UN ISO 3166-1 alpha-3 codes, i.e., Australia (AUS), Brazil (BRA), Barbados (BRB), Central African Republic (CAF), China, mainland (CHN), Falkland Islands (FLK), Indonesia (IDN), India (IND), Mongolia (MNG), New Zealand (NZL), South Sudan (SSD), USSR (SUN), United States of America (USA), and Uruguay (URY).

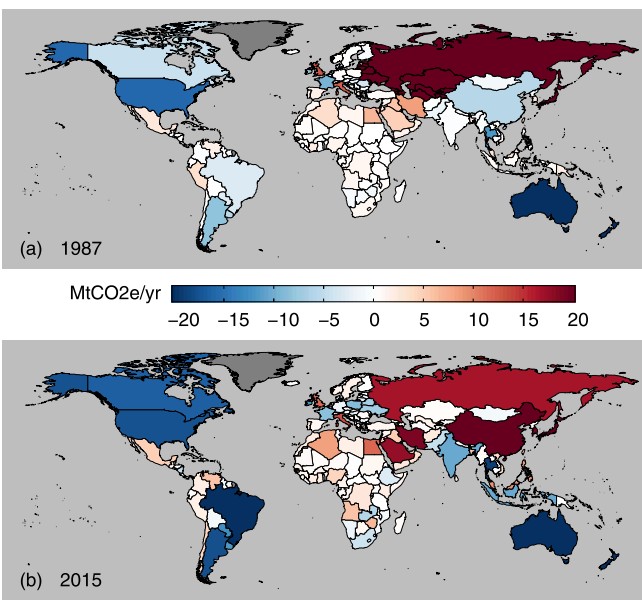

**Fig. 2 Differences between trade-adjusted agricultural emissions (TAE) and production-based emissions (PBE).** Differences are shown for 1987 (a) and 2015 (b), in total absolute terms.. These differences are calculated by subtracting PBE from TAE, such that the larger the positive difference (darker red), the higher is a country's TAE compared to its PBE (i.e., net importers of agricultural emissions). Likewise, the larger the negative difference (darker blue), the lower is a country's TAE than its PBE (i.e., net exporters of agricultural emissions). Dark grey colours indicate countries with no available data (e.g., Greenland). Per capita differences are shown in Supplementary Fig. 5.

East[28]. Specifically, for countries like Bahrain, Kuwait, and the United Arab Emirates, TAEs were at least four times larger than their respective PBEs in 2015. In mainland China, TAE was only around 7.9% higher than PBE in 2015, but this was equivalent to a difference of 51.5 Mt $CO_{2e}$/yr – the largest gap in 2015 for countries with larger TAE than PBE in total absolute terms.

However, in per capita terms, this difference for mainland China was relatively small (<0.05 t $CO_{2e}$/cap/yr in 2015).

Meanwhile, major exporters had lower TAE as compared to PBE. We found, for example, that countries in Oceania such as Australia and New Zealand had an average TAE that was 59.3% lower than PBE in 2015. Additionally, some major exporters saw a rising gap between PBE and TAE from 1987 to 2015. In Brazil, for example, PBE in total absolute terms was only 3.08 Mt $CO_{2e}$/yr higher than TAE in 1987, but this gap widened by 45.6 Mt $CO_{2e}$/yr in 2015. This trend in Brazil demonstrates the rapid growth in export-driven production emissions in the country with increasing trade[29]. For Brazil, the export emissions of ruminant meat tripled from 10.007 Mt $CO_{2e}$/yr in 1987 to 34.453 Mt $CO_{2e}$/yr in 2015. These export emissions were much larger than those of the second largest contributor to Brazilian export emissions, i.e., 'others', of which soybeans were the dominant items. Although soybean exports were larger than those of ruminant meat during both periods, the entire 'others' food group to which soybeans belong only contributed 1.229 Mt $CO_{2e}$/yr and 12.727 Mt $CO_{2e}$/yr in 1987 and 2015 respectively, because of a lower emission intensity than ruminant meat.

**Emissions embodied in trade flows.** We found that global emissions embodied in food exports rose from 306.5 Mt $CO_{2e}$/yr in 1987 to 695.7 Mt $CO_{2e}$/yr in 2015, while for food imports, embodied emissions rose from 317.0 Mt $CO_{2e}$/yr in 1987 to 724.5 Mt $CO_{2e}$/yr in 2015. Theoretically, both export and import emissions should be equal at the global level[13], but because of bilateral trade number inconsistencies in the original FAOSTAT datasets, such differences are to be expected[30].

At the regional level, the trend was largely similar (Fig. 3). Throughout the period 1986–2017, Europe had the largest emissions embodied in both exports and imports, including intra-regional trade. The regions Latin America and the Caribbean and Asia saw a larger growth in export and import emissions respectively, with the latter region's import emissions surpassing those of Europe in the last two decades. These trends

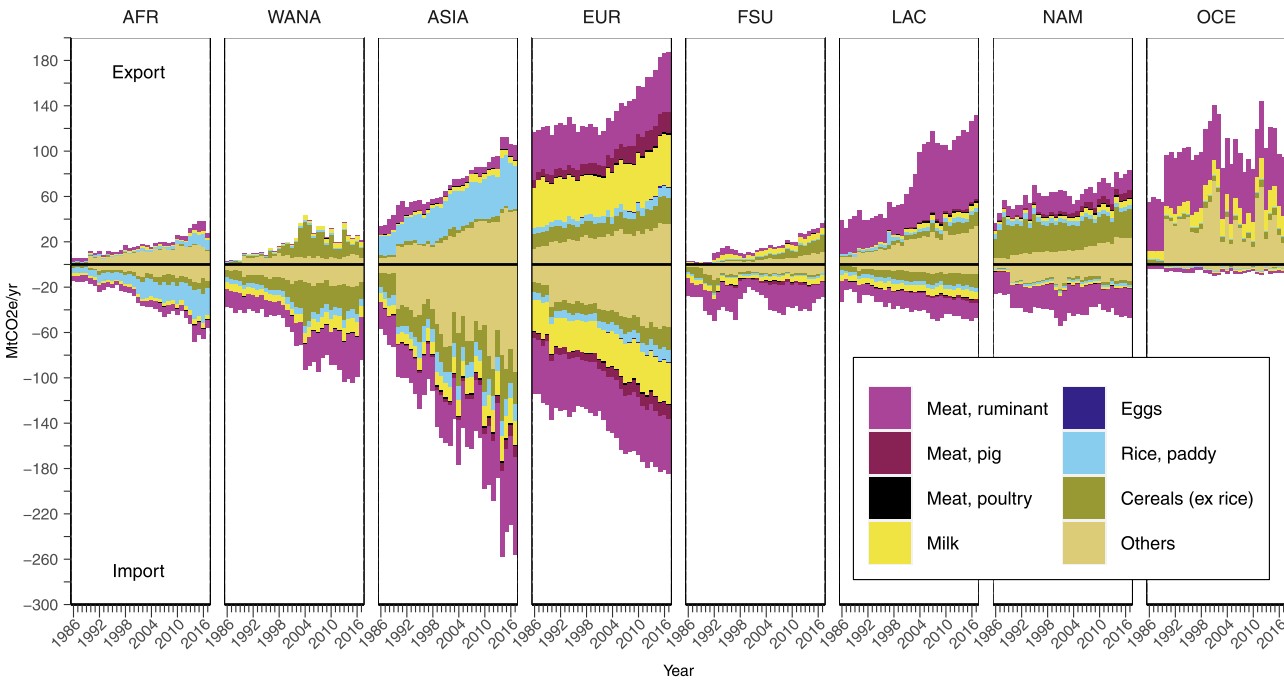

**Fig. 3 Regional changes in emissions embodied in food exports and imports from 1986 to 2017.** Emissions embodied in exports are shown at the top half of the figure, while imports are shown at the bottom half. The regions analysed are Africa (AFR), Western Asia and Northern Africa (WANA), Asia (ASIA), Europe (EUR), Former Soviet Union (FSU), Latin America and the Caribbean (LAC), North America (NAM), and Oceania (OCE). Additionally, the figure also provides a breakdown of the contribution of each food group. Values are shown for individual years, i.e., without averaging values for three years.

mirror the changes in trade volumes, with an increase in volumes being associated with rising emissions embodied in trade and vice versa (Supplementary Fig. 6).

To understand where emissions were exported to or imported from, we estimated the proportions of intra- and inter-regional emissions embodied in trade flows for each region (Fig. 4). In 2015, a large share of emissions embodied in exports and imports in Europe were intra-regional (80.3% and 76.9% respectively), whereas regions, such as Africa, Western Asia and North Africa, the Former Soviet Union and Asia had a larger share of inter-regional imported emissions (66.7–85.8%). These trends clearly reflect regional trade patterns and policies, such as the pro intra-regional trade policies of the European Union, and the rise in affluence, trade liberalisation and import dependence of Asian economies[31].

For Oceania, despite the low volumes of food exported internationally, the region contributed to a large proportion of imported emissions to regions such as Asia and North America (more than 27% on average in 2015 for both regions). Based on our data, two reasons explain this: (1) Oceania exported a high proportion of more emission-intensive ruminant meat to these regions; (2) the agricultural emission intensity of the food group 'others' in Oceania was higher than the intensity of 'others' of all other regions exporting to Asia and North America in the period of 2014–2016.

Another similar example is that of imports to Africa in 2015, in which Asia was the largest exporter to Africa in terms of volume. However, in terms of emissions, Asia was only second to African intra-regional imports. Two reasons might explain this: (1) rice was the main contributor to Africa's import emissions, and (2) Africa's average emission intensity of paddy rice (1.14 kg $CO_{2e}$/kg product) was higher than that of Asia (0.90 kg $CO_{2e}$/kg product). Hence, this suggests that a country's or region's relatively low or high volumes in trade may under- or overestimate its contribution to traded emission flows.

**Food groups**. In our analyses of TAEs and emissions embodied in trade (see Figs. 1 and 3), we found that ruminant meat followed by milk products had the largest share of embodied emissions for most regions, particularly in Europe, Oceania and North America. In Asia, paddy rice was a more dominant contributor to TAEs and export emissions, owing to the importance of rice cultivation in the region.

In more recent years, however, the food group 'others' contributed to an increasing share of TAEs and emissions embodied in trade for all regions. This follows the growing demand for crop-based feed in recent years to meet the rising demand for animal-source food associated with changes in dietary pattern[32]. For example, soybeans, which fall under our 'others' food group, saw a fivefold increase in global exports from 1986 to 2017.

**Alternative specifications**. Our three sensitivity analyses demonstrate the robustness of our TAE estimates (Supplementary Figs. 7, 8, 9, 10). In the first analysis, we used global emission intensities to calculate TAE where a non-producer country is concerned, instead of using regional emission intensities which was the case for the original analysis (see Methods - Sensitivity analyses). By switching to global emission intensities, we thus assumed a more global, inter-connected food trade network where imports are not restricted to a country's region. We found that the switch to global emission intensities only produced a very small difference during the entire period of 1986–2017: for countries that saw an increase in TAE from the original approach, the mean change was +1.5%; for countries that saw a decrease in TAE, the mean change was −0.47% (Supplementary Fig. 7). The largest differences were consistently found in several small-island and import-dependent countries where differences in emission intensities are more apparent in the resulting TAEs due to the larger share of traded products in their inventories. This was the

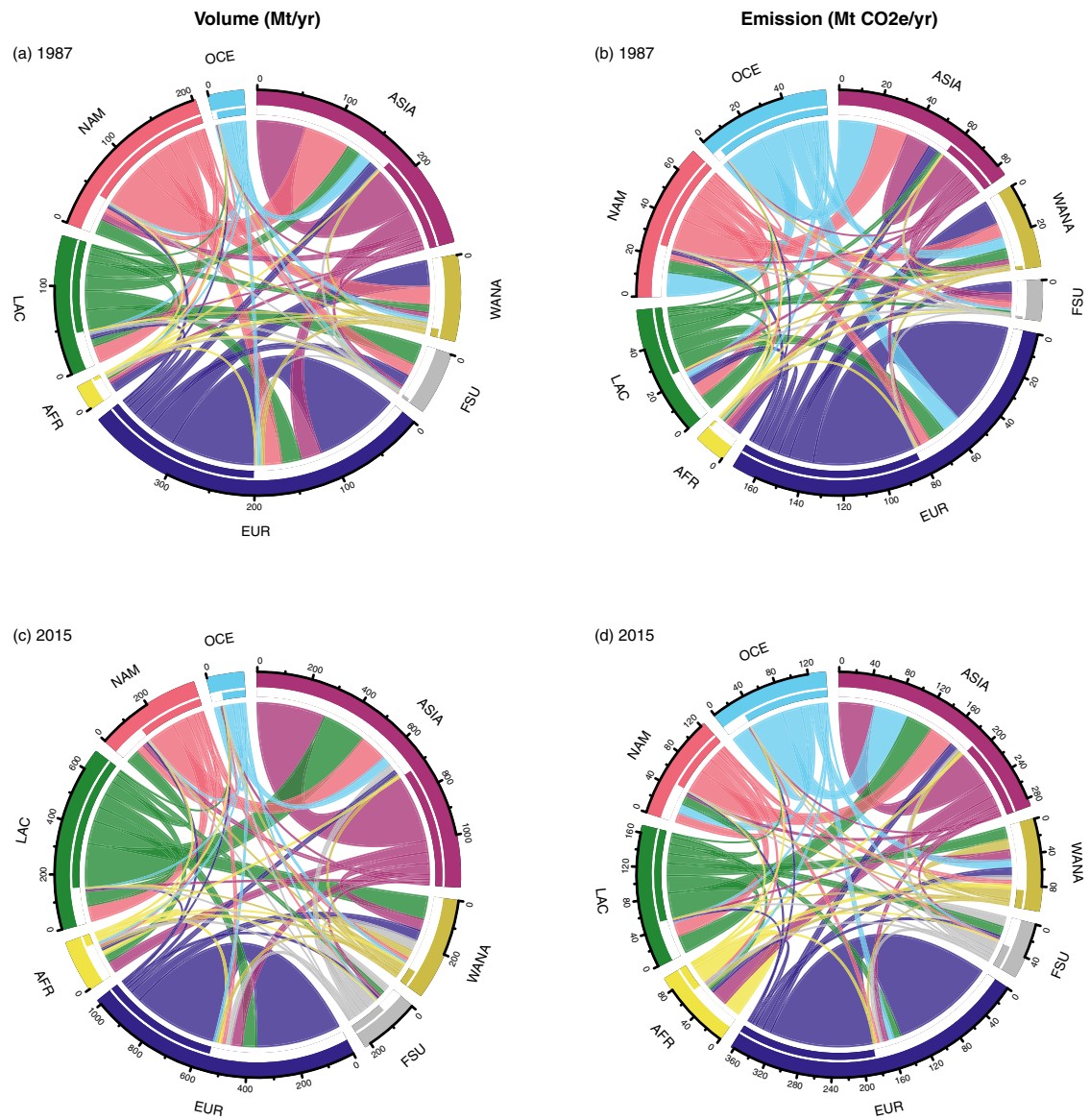

**Fig. 4 Inter- and intra-regional trade flows in 1987 and 2015.** Trade flows in terms of volumes (Mt) are shown on the left (**a**, **c**), while trade flows in terms of embodied emissions (Mt CO$_{2e}$/yr) are shown on the right (**b**, **d**). The first coloured bar from the outside of each circle represents the sum of both exports and imports, whereas the second coloured bar indicates exports from the region's countries. Colours of the trade flows correspond to the exporting region. The regions analysed are Africa (AFR), Western Asia and Northern Africa (WANA), Asia (ASIA), Europe (EUR), Former Soviet Union (FSU), Latin America and the Caribbean (LAC), North America (NAM), and Oceania (OCE).

case for Kiribati, which saw the largest difference of almost 19 times in 2000. These extreme values, however, only accounted for a small share of differences during the period of 1986–2017, with over 98% of differences falling within +/−10% from the original approach. Our approach of using regional emission intensities was therefore sufficient in capturing the wide-reaching nature of global food trade.

In the second analysis, we adopted a technology-adjusted approach in calculating TAE to account for technological differences in carbon efficiencies in the exports of different countries[20]. This approach has been proposed as an improvement to the conventional consumption-based accounting by more accurately reflecting how changes to national climate policies could affect overall global emissions[20]. Similar to the first analysis, the difference in switching to this approach was minor during the period of 1986–2017: for countries that saw an increase in TAE from the original approach, the mean change was +5.5%; for

countries that saw a decrease in TAE, the mean change was −8.5% (Supplementary Fig. 8). The largest difference was found for the United Arab Emirates, where TAE deviated by as much as 60 times from the original approach in 1997. In addition to being highly trade-dependent, this large difference also coincided with the United Arab Emirates having very different emission intensities from the global average: for example, emission intensities for cereals were as much as 90 times higher than the global average in the same period. Nevertheless, these extreme values only made up for a small share of differences during the period of 1986–2017, with over 81% of differences falling within +/−10% from the original approach. Our results were therefore relatively similar when comparing to a technology-adjusted approach. However, this approach did produce some interesting patterns at the regional and food group-specific levels (Supplementary Figs. 8, 10). For example, Europe and North America consistently had lower TAEs for ruminant meat under

this approach, reflecting the higher efficiency of ruminant meat production in these regions[21].

In the third analysis, we used emission intensities that account for both domestic production and imports, instead of only using domestic production-based emission intensities which was the case in the original approach. In this sensitivity analysis, we therefore tested the sensitivity of our approach if potential re-exports were accounted for. As was the case with the first and second sensitivity analyses, using this re-exporter approach also produced very small differences in most cases. During the period of 1986–2017, we found that, for countries that saw an increase in TAE from the original approach, the mean change was +1.3%; for countries that saw a decrease in TAE, the mean change was −5.4% (Supplementary Fig. 9). Similar to the second sensitivity analysis, the largest differences were found for the United Arab Emirates, with TAEs deviating by as much as 62 times from the original approach in 1997. Looking at specific food groups, we also found large differences among countries in Oceania, the Middle East, southern Africa, and parts of Europe (Supplementary Fig. 9). As an example, for paddy rice in 2015, the Netherlands and Slovenia saw deviations in TAE by almost 5 and 6 times respectively when using the re-exporter approach. These countries re-exported around 60% of their paddy rice imports in 2015, resulting in these deviations[33]. However, we observed such differences in only a few cases, mainly for intermediary trading countries, and over 91% of differences fell within + / −10% from the original approach during the period of 1986–2017. Hence, the results of our main approach highlight the role of intermediary trading countries in reducing emissions, especially if their exporting partners have higher emission intensities than theirs. This is important considering that approaches that account for re-exports allocate all emissions to countries where consumption takes place, ignoring the potential role of intermediary trading countries in terms of choosing exporting partners with lower emission intensities.

**Land-use emissions**. In addition to our main focus on farm-gate emissions, we also tested how agricultural land-use emissions, using FAOSTAT land-use emissions data[34], would change when adopting a trade-adjustment approach for the year 2015 (see Methods).

We found that several major agricultural exporters had relatively high agricultural land-use emissions, as well as large decreases in such emissions when adjusted for trade flows (Fig. 5; see also Supplementary Fig. 11 for per capita results). Indonesia's

agricultural land-use emissions, for example, decreased by 471.9 Mt $CO_{2e}$/yr when adjusted for trade flows, representing more than 40% of its original agricultural land-use emissions. In Brazil, this decrease after adjusting for trade flows was 120.1 Mt $CO_{2e}$/yr, which was more than 20% of its original agricultural land-use emissions.

Conversely, some countries with relatively low agricultural land use emissions saw a large increase in emissions when adjusted for trade flows (Fig. 5; Supplementary Fig. 11). For example, the agricultural land use emissions of mainland China and India were only 2.51 Mt $CO_{2e}$/yr and 9.32 Mt $CO_{2e}$/yr respectively, but these figures increased by a factor of more than 50 and 10 respectively when trade flows were included. We also saw similar results for a number of import-dependent countries, such as several European countries (e.g., Italy and Spain, where the increase was more than 60 times, the highest increase among all countries in Europe), and small island states (e.g., Singapore, where agricultural land use emission was less than 0.1 Mt $CO_{2e}$/yr, but the increase was more than 100 times with trade adjustments). These results reflect the export- and import-driven nature of land-use emissions in many countries, particularly where agricultural land-use emissions are high, such as Brazil and Indonesia, or where there is a high dependency on agricultural imports, such as mainland China and Europe.

## Discussion

Our study adds further insights to on-going discussions over emissions accounting with a focus on food and the agricultural sector. Specifically, we calculated emissions using a trade-adjustment approach that utilises bilateral trade data as reported in FAOSTAT. As such, we shift the focus beyond the producer and consumer ends of the supply chain, and bring into discussion the role of intermediary trading countries such as those in the Middle East and Europe in reducing emissions. Additionally, considering the differential roles of producer, consumer, and intermediary trading countries in reducing emissions, we tested our approach using alternative specifications as shown in our three sensitivity analyses. By introducing our emissions accounting approach into the debate, we therefore extend the findings of previous studies that have also used trade-adjustment approaches to estimate emissions[16–19,35].

Importantly, as evident from the results of the third sensitivity analysis, the estimates from our main approach did not deviate substantially from alternative specifications that accounted for re-exports. This is further substantiated when comparing our results

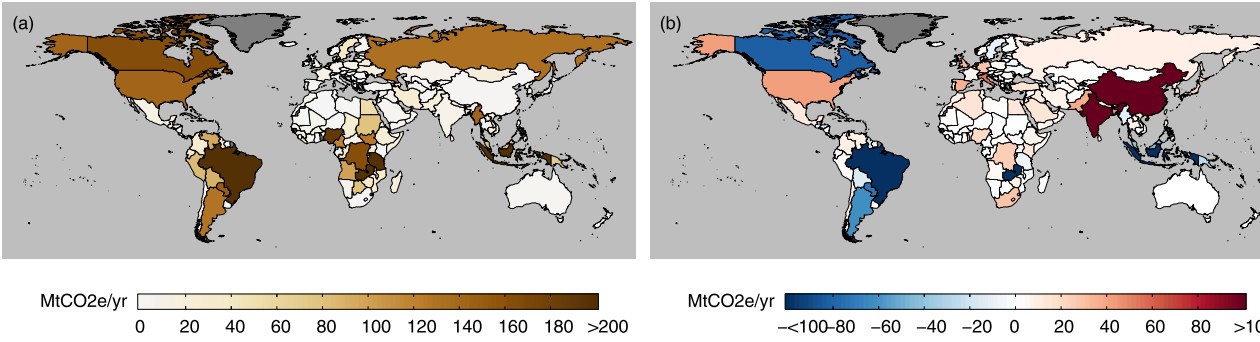

**Fig. 5 Agricultural land-use emissions for 2015.** Left panel (**a**) shows agricultural land use emissions without trade adjustments. Right panel (**b**) shows the differences between trade-adjusted agricultural land-use emissions and agricultural land-use emissions. Differences are calculated such that the larger the positive difference (darker red), the higher is a country's trade-adjusted agricultural land use emissions (i.e., net importers of land-use emissions). Likewise, the larger the negative difference (darker blue), the lower is a country's trade-adjusted agricultural land use emissions (i.e., net exporters of land-use emissions). Dark grey colours indicate countries with no available data (e.g., Greenland). Per capita emissions and differences are shown in Supplementary Fig. 11.

with those of other studies that have accounted for re-exports using different approaches. For example, when comparing our TAE results with those of Romanello and colleagues[18], whose geographical and temporal scopes closely resemble ours, we find that the trends for all regions are largely similar (see Supplementary Fig. 12). While the differences are quite noticeable in relative terms, particularly in the Former Soviet Union and Oceania (see Supplementary Fig. 15), it should be noted that the differences are already evident when looking at production-based emissions (i.e., before trade adjustments; see Supplementary Figs. 13–14), suggesting that these discrepancies are not solely due to methodological differences in accounting for re-exports, but instead could simply be a case of using different emission factors and food groupings. Furthermore, our results shed light on how agricultural emissions travel across the global trade network of food, and how these relate not only to trade volumes, but also to the relative emission intensities between trading partners. Therefore, estimating a country's embodied emissions by trade volumes alone while ignoring the differences in emission intensities can under- (or over-) estimate its underlying embodied emissions. These differences are especially relevant for consumptions and trade flows that involve a large share of emission-intensive items such as ruminant meat, milk products, and rice.

Additionally, our findings challenge the general notion that more economically advanced countries tend to have higher levels of emissions embodied in per capita consumption and imports[9,13,36]. Our results show that this is not always the case when considering emissions within the farm-gate, as several low-income or emerging and developing economies also display some of the highest per capita TAEs. Rather, these high TAEs have a lot to do with the consumption of emission-intensive food items, the employment of less efficient agricultural practices, or simply being a major intermediary trading country.

An important aspect of our study is to explore how trade adjustments to emissions accounting could differ from the conventional production-based approach. We show that the differences are particularly noticeable for major agricultural importers and exporters. While this difference has already been discussed in other studies for all sectors collectively[7,9,11–14], our findings go one step deeper by exploring these differences specifically within the agricultural sector. These differences between the production-based and trade-adjusted approaches also highlight the technical issues surrounding nationally determined contributions (NDCs) under the Paris Agreement and their effectiveness in mitigating emissions. As it stands, many of the targets and plans described in countries' NDCs are national in scope and include policies and measures to reduce non-$CO_2$ GHG emissions from agriculture[37], but they do not address the role of trade in displacing agricultural emissions. Thus, our findings support the call for GHG emission inventories to consider adopting a consumption-based approach that considers the various implications of trade, which is still largely missing in national-level food-related emissions accounting[4].

Concurrently, our results also clearly show the major contribution of animal-source food, particularly of ruminant meat and milk products, in overall emissions embodied in consumption and trade in general. Thus, by extension, our study supports and justifies the growing call to consume healthy diets consisting of a low amount of emission-intensive meat and dairy products, to achieve climate change mitigation targets more effectively[3,16,24,38–40].

The results of our study are inevitably bound by the limitations of our main dataset, i.e., FAOSTAT. One limitation is the lack of specific emission intensities for a number of items in the FAOSTAT databases such as fruits, oilcrops, and vegetables, all of which we categorised under the food group 'others' (see Supplementary Table 2), and to which we assigned a single emission intensity (see Methods). Although emission intensities can vary

considerably depending on the level of detail used to estimate them, this variability is less of an issue for our analysis in which country-level emissions are estimated by both intensities and volumes[41]. Furthermore, since these 'other' items are responsible for only 14.3% of total agricultural production emissions from 1986 to 2017, our study has already captured a majority of emissions based on food group-specific emission intensities.

Furthermore, our study is mostly focused on emissions from the agricultural production stage, and does not consider emissions from other pre- and post-production processes. One notable process missing from our main TAE estimates is emissions from FOLU and land-use change, which are key contributors to food system emissions alongside production stage processes[5,42,43]. We attempted to address this gap by testing how agricultural emissions from FOLU would change when adjusted for trade flows. To do this, we assumed that all food items drive land-use emissions proportionally to their production volumes due to the recognised limitations in linking FOLU emissions to specific agricultural items. Specifically, this is because of the difficulties in establishing a cause-and-effect mechanism between food items and land-use change in a consistent manner[23,44,45]. Moreover, our analysis implicitly accounted for emissions due to the production of crops that are used as feed items, as several crops such as soybeans can be used both directly for human food consumption and as livestock feed (see Supplementary Table 2).

Likewise, our study did not account for transportation emissions. Although food transport reduction through the regionalisation of food systems is an important climate change mitigation option[46], transport constitutes a very small share of overall food system emissions[5,6,42,47]. Therefore, excluding its emissions would not make much difference in our results. Nevertheless, bringing producers and consumers closer through regionalised food systems could lead to more responsible food production and consumption[48], thus future research could also consider transportation emissions as an extension of our findings.

Interpretation of our findings also warrants some understanding of the different ways in which emissions embodied in the consumption of food and agricultural products can be estimated. These include: (1) the multi-regional input-output (MRIO) approach that explores trade for final consumption, and (2) the BTIO approach, which we applied in this study[49,50]. Although the MRIO approach more accurately captures the concept of a consumption-based analysis, the BTIO approach is more transparent and suited for analysing bilateral political agreements as well as trade and climate policies[12]. This is because it correlates directly with monetary bilateral trade data, and considers bilateral trade without splitting it into intermediate and final consumption[12]. Furthermore, the BTIO approach is also preferable for assessing border tax adjustments[51], and is increasingly being used to test the pollution haven hypothesis[49].

With these characteristics of the BTIO approach, we believe that our study provides a foundation for further research that seeks to investigate the links between agricultural emission inventories, trade patterns, and international climate policies. For example, because our method is consistent with bilateral trade data, our results could be used to examine the effects of trade agreements and tariffs on emissions embodied in trade and overall emission inventories, and vice versa. This application would be an extension of the findings of previous research that have looked at the trade creation and diversion effects, as well as border effects of trade agreements, specifically for the agricultural sector[52–54].

Another possible follow-up from our study is to calculate embodied emissions in terms of nutrients. This would be an interesting avenue to investigate, as studies have suggested that emission estimations could vary when switching from volume- to

nutrient-based accounting[55,56]. Future studies linking emissions and nutrients embodied in trade could also explore how efforts to reduce emissions via trade adjustments could impact the distribution of nutrients, as international food trade is essential for nutrient access, particularly for less economically well-off countries[57].

As other similar approaches do, our consideration of TAEs in national emission inventories for the agricultural sector acknowledges demand as a driver for emissions. Additionally, our focus on using bilateral trade data ensures that trading partners, i.e., countries simply functioning as intermediaries, and not just producers and final consumers, are also included in emissions accounting. This is equally important as trade also benefits those who are intermediary in the global supply chain. Under current climate change mitigation frameworks, national or regional commitments focus on production-based emissions, with demand playing only a minor role. As a result, this could lead to a displacement of emissions overseas. For example, stricter emission reduction targets for the European Union have been found to lead to a considerable displacement of saved emissions abroad, where agricultural emission intensities are higher[58]. Our analysis in calculating trade-adjusted emissions in the agricultural sector would have significant and practically relevant trade policy implications, and could serve as an important element towards constructing trade-adjusted national or regional emission reduction targets. In combination with trade policy regulations, such as border adjustment measures as recently proposed by the European Union for emission-intensive commodities like steel, cement, aluminium and fertilizers, such trade-adjustments would provide incentives for producer, consumer, and intermediary trading countries to consider emissions intensities in agricultural production and trade, consequently leading to more cost-effective emission reductions overall.

## Methods

**Data and preprocessing**. We used data on agricultural production, trade and emissions available from FAOSTAT as of 02.12.2019[33]. The data is available for 211 countries, which we grouped into eight regions based on the UN M49 Standard Geographical Classification to also have a regional overview (see Supplementary Table 1)[59].

The trade data consist of primary and processed items (443 items in total) and are available from two datasets: the Trade – Crops and Livestock (TCL) dataset, which provides the total export-import figures for each country and year, and the Detailed Trade Matrix (DTM) dataset, which provides a breakdown of partner countries in bilateral trade flows as reporter countries (i.e., countries that report directly to the FAO or other statistical bodies) and partner countries (i.e., countries which reporter countries record as their trading partners). We restricted the trade data to food-related items that are consumable as human food or are by-products of items cultivated for food (307 items; see Supplementary Table 2).

Our study mainly used trade figures of reporter countries from the DTM dataset because of the higher level of details provided in terms of trading partners. Hence, our study's time frame was limited to the period of 1986–2017, which is the time frame available in the DTM dataset. However, when a country is not listed as a reporter country in the DTM, we filled the data gaps by taking the trade figures from all other reporter countries that list the respective country as a partner country. Also, we found discrepancies in the total trade figures between both the DTM and TCL. Thus, to maintain consistency in trade numbers, we scaled trade figures in the DTM to match those recorded in the TCL (Supplementary Fig. 16).

To estimate emissions of the traded items, we used agricultural emission intensities provided by FAOSTAT[60], which are computed using the IPCC Second Assessment Report's global warming potentials[61]. However, because these emission intensities are provided for primary items only, and as the trade datasets also contain processed items, we initially converted the mass of processed items ($m_f$) to their primary equivalent mass ($m_{eq}$). We did this by adopting the approach used by Kastner et al. (2011), in which the ratio of the caloric contents of both processed ($C_f$) and primary ($C_i$) items is used as the basis for conversion (Eq. (1))[62]. We obtained caloric content values (in kcal/100 g) from the following sources: FAO and USDA[63,64].

$$m_{eq} = m_f * \frac{c_f}{c_i} \qquad (1)$$

FAOSTAT provides emissions and emission intensity data for only 14 food groups, namely cereals, rice and several animal products[65], which cover 82.3 –

92.5% of total agricultural emissions from 1986 to 2017. We added another food group called 'others' to include items not originally covered by FAOSTAT emission groupings (e.g., vegetables, fruits, and oilcrops). To calculate the emission intensities of 'others' at the country, regional and global levels, we utilised FAOSTAT's total agricultural emissions data[66], and proceeded with the following method: we first calculated the total emissions from 'others' by subtracting agricultural emissions by the emissions from the 14 groups for each country and year. Afterwards, we obtained the emission intensity of 'others' by dividing the calculated emissions with the production volumes of these items.

In addition to using FAOSTAT agricultural emission intensities, we also estimated an alternative set of emission intensities based on agricultural land-use emissions. We did this by using FAOSTAT agricultural land use emissions data, based on the IPCC Fifth Assessment Report's global warming potential coefficients[34]. These emissions include processes related to net forest conversion, degradation of organic soils, and burning of biomass in humid tropical forests and peat soils[67] – processes that are distinct from those covered in FAOSTAT agricultural emission intensities[65]. To calculate agricultural land use emission intensity, we divided land-use emissions by the production volumes of all items for each country and year. Due to the lack of information on the contribution of each item to land-use emissions, we assumed that all items contributed equally to emissions, hence each item was assigned the same emission intensity.

**Bilateral trade input-output (BTIO) approach**. To calculate trade-adjusted agricultural emissions, we adopted a BTIO approach, also known as the emissions embodied in bilateral trade approach[12,49,50]. Using this approach, we did not distinguish final and intermediate trade flows when calculating a country's agricultural emissions, but rather we exogenously include imports and exports so that our estimations correlate with bilateral trade data as they appear in FAOSTAT[49,50].

Our BTIO approach is based on the following assumptions: (1) for an exporter country, we assumed that the exports originated purely from the country's own domestic production, in order to tie emission intensities directly to bilateral trading partners. If the exporter country did not produce the export item (i.e., a non-producer country), we (2) assumed that the exporter would have imported the item from another country within the same region, thus assuming that countries have a preference to import items from geographically closer trading partners. If the region did not produce the item, only then did we assume that the item was imported from elsewhere globally. The implications of switching to global emission intensities are explored in the first sensitivity analysis (see Sensitivity analyses). These assumptions are important when considering the types of emission intensities to use to calculate emissions embodied in trade.

**Embodied emissions**. We calculated emissions ($E$) by multiplying the volume of the activity of interest (in tonnes of primary equivalent mass $m_{eq}$) with the emission intensity ($EI$) of an item $i$ for a country $c$ in year $y$ (Eq. (2)). Activity in this case refers to agricultural production, export, and import of commodities:

$$E_{c,y,i} = m_{eq\,c,y,i} * EI_{c,y,i} \qquad (2)$$

We define trade-adjusted agricultural emissions (TAE) as the sum of production-based emissions (PBE) and import emissions (ImE) minus export emissions (ExE)[13], as shown in Eq. (3).

$$TAE_{c,y,i} = PBE_{c,y,i} + ImE_{c,y,i} - ExE_{c,y,i} \qquad (3)$$

Throughout this study, values are shown in terms of three-year averages, unless specified otherwise. This practice of averaging values from multiple years (usually three or five years) to represent single year values is commonly used to keep figures comprehensive and representative of wider time frames[17,43].

**Sensitivity analyses**. In our first sensitivity analysis, we relax assumption (2) by using global average emission intensities instead of regional ones to calculate TAEs where a non-producer country is concerned. To conduct this analysis, we used global average emission intensities to calculate ExE when estimating the TAE of a non-producer exporter country. For an importer country that is importing from a non-producer country, we used the global average emission intensity to calculate ImE when estimating the importer country's TAE (Supplementary Fig. 16).

The second sensitivity analysis adopted a technology-adjusted approach based on the methodology by Kander et al. (2015) which accounts for technological differences in carbon efficiencies in exports, thus more accurately reflecting how changes to national climate policies could affect global emissions[20]. For this analysis, TAE was estimated using the same approach as the original method, with one exception: we used the global average emission intensity to calculate ExE when estimating the TAE of an exporter country, regardless if the country is a producer or non-producer. The reason for doing so is that, if we are to estimate the effects of a country's exports on global emissions, we also need to know what alternative productions it could replace – but because of the lack of knowledge of what would be the substitute, we use the global average as a plausible assumption[20].

In the third sensitivity analysis, we relax assumption (1) by estimating TAEs using emission intensities that account for both production and imports. To do this, we calculated, for each country, the sum of emissions from domestic production and imports, using the relevant emission intensities as provided by FAOSTAT. We then divided emissions with the sum of production and import

volumes to obtain a new set of emission intensities. Using this approach, we were able to link country-specific production emissions with imports in each trade flow. This essentially differs from assumption (1) of the original analysis, as we now consider that countries export both domestically produced and imported items. Under this relaxed assumption, exports come in proportional shares from import and domestic production, due to the lack of detailed information on the shares that go into exports[62]. Furthermore, the new set of emission intensities considers first-order trade (i.e., exports of partner countries' production), rather than second-order trade and beyond (i.e., exports of partner countries' imports).

For all sensitivity analyses, we calculated TAE and emissions embodied in trade flows as before. We then estimated the weighted difference ($E_{diff}$) between these results ($E_{SA}$) and those of the original method ($E_O$), as shown in Eq. (4).

$$E_{diff} = \frac{E_{SA} - E_O}{E_O} \qquad (4)$$

**Reporting summary**. Further information on research design is available in the Nature Research Reporting Summary linked to this article.

## Data availability
All processed data, country data, and food data are available in the Figshare database under the following link: https://doi.org/10.6084/m9.figshare.19583194. The original FAOSTAT data are available in the FAOSTAT open-source database under the following link: https://www.fao.org/faostat/en/#data. The data from Romanello et al. (2021) was obtained upon request from the authors; visualisations are provided in the report's data explorer platform: https://www.lancetcountdown.org/data-platform/mitigation-actions-and-health-co-benefits/3-5-food-agriculture-and-health/3-5-1-emissions-from-agricultural-production-and-consumption. Food caloric content values were obtained from FAO (2019): http://www.fao.org/economic/the-statistics-division-ess/publications-studies/publications/nutritive-factors/en/; and USDA (2018): https://fdc.nal.usda.gov/index.html/.

## Code availability
The code used for this study was generated using R version 3.6.3 (2020-02-29), and RStudio (Version 1.2.5033). Codes are available in the Figshare database under the following link: https://doi.org/10.6084/m9.figshare.19583194.

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

## Acknowledgements

The authors are sincerely grateful to Harry Kennard and Carole Dalin for their insightful comments and support in providing the raw data that was used for the paper by Romanello et al. (2021). The data enabled the authors to compare their results with those of Romanello et al. (2021) and to produce Supplementary Figs. 12–15 in the Supplementary Information. P. Pradhan acknowledges funding from the German Federal Ministry of Education and Research (BMBF) for the SUSFOOD project (grant agreement No 01DP17035) and for the BIO-CLIMAPATHS project (grant agreement No. 01LS1906A) under the JPI Climate AXIS ERA-NET. The funders had no role in the design, data collection and analysis, decision to publish, or preparation of the study. The data used is listed in the references.

## Author contributions

A.F., P.P., O.F. and J.P.K. contributed to the conceptualisation, writing and editing of the paper. A.F. conducted the analysis. P.P. and O.F. supervised the research.

## Competing interests

The authors declare no competing interests.
