## [Peer Review File · Nature Communications]

Reviewer comments, first round review –

Reviewer #1 (Remarks to the Author):

Dear authors,

Please, see below my reviewer report on your manuscript "Consumption-based greenhouse gas emissions of the agriculture sector":

Revision of NCOMMS-20-24771-T: Consumption-based greenhouse gas emissions of the agriculture sector

The manuscript by Foong, Pradhan and co-authors has been now revised for potential publication in Nature Communications.

The study is, mainly due to the topic addressed, timely and of great interest for a broad audience. Authors assessed the demand-side GHG emissions in a global context, with a disaggregation of information in terms of regions and food items. A global assessment on demand-side emissions associated to food consumption worldwide is undoubtedly attractive and of great scientific and social interest.

One of the main outcomes of this research is the insight on the importance of being an agricultural-product exporter vs an importer country over the weight of consumption vs production GHG related emissions on the countries' GHG share. This is well exemplified by authors with a scenario approach.

The work is fully based on FAO statistics which, in principle, should make results robust and reliable. However, there are some major methodological constraints I would like to point out as they could prevent the publication of this manuscript in its current form. Some of these limitations were even recognized by authors.

1. Apparently, authors did not include land use change related GHG emissions in their estimations. I see this as a substantial methodological limitation that might impact on the final figures of GHG emissions, of both production and consumption. This is mentioned by authors in the following statement:

“Finally, the findings of our study are based on emissions data for agricultural production only, and we did not account for other stages of food systems such as land use change and post-production processes due to the lack of comprehensive data covering these other emission sources in the FAOSTAT databases.”

As said, this is highly important and could substantially change their findings for (e.g.) consumption emissions of imports from Latin America (such as EU).

Emissions from pre-production processes (e.g. inputs production) and transport are not included in

the analysis and this is a methodological disadvantage comparing with “other input-output data” authors referred in line 271.

2. The following statement (lines 263-268): “We further postulate that tracing the precise pathways of exported items may leave re-exporters out of the emissions accounting picture altogether. From an ethical point of view, we believe that such re-exporters should also bear some responsibility to emissions embodied in global trade and consumption, as they ultimately benefit from such trade economically despite not consuming the items themselves.” Took me to the following considerations:

- Bilateral trade input-output (point 2 of Methods section) has a very specific meaning, which comes from a still open discussion regarding to both methodological and policy-analysis aspects. This should be mentioned somehow in the manuscript. On the contrary, statements such as the above mentioned could be understood as a way to somehow avoid recognition to the Multi-Regional Input-Output (MRIO) allocation approach, as well as other approaches that deal with this issue (e.g. "shared responsibility", see Lenzen et al. 2007).
- If authors address emissions from consumption, they should stick to that. If authors want to attribute impacts and responsibilities along the value chain, they should carry out a specific analysis of that, studying the share of emissions and the economic value that corresponds to each stage of the chain.
- The concept of “Consumption-based approach” (CBA), pivotal for this research (appearing even in the title), is of great interest. As pointed out by authors, it (undoubtedly) should be generalized in the field of GHG mitigation within the agro-food sector. However, when authors talk about CBA, the focus should be on final consumption so re-exporters would be less important. Of course, re-exporter and intermediates have their responsibility in GHG emissions (and mitigation) but there are appropriate indicators to do so and "bilateral trade input-output" could be a methodological option to address this issue.

The CBA concept and approach has been explored before by several authors. Peters et al. (2008) already stated its importance due to potential "carbon leakage". In the input-output (IO) community, researchers firstly worked with available national input-output tables. Then, authors utilized global Multi-Regional Input-Output (MRIO) modelling tools (e.g. Eora, Exiobase, Fabio, etc.). The different tools and approaches used led to different notions of consumers' footprints, depending on the IO models used and how bilateral trade is conceived and used. In this context, Kanemoto et al. (2012), Su & Ang. (2011) and Cadarso et al. (2018) worked deeply on these different perspectives. These authors distinguished two approaches: bilateral trade input-output (BTIO) and MRIO. This basically means that there are two ways of calculating the CBA (or consumers' footprint) from an IO perspective. Each of them based on different allocation procedures for products in an international trade. As said, this discussion is still open and beyond methodological aspects (e.g. what to do with re-exporters), both approaches are useful. The advantage of MRIO is that it is closer of CBA as emissions are assigned to a final consumer (no matter of the number of borders crossed by a certain good). Piñero et al. (2020) considered all this but focusing on raw materials instead of emissions (but their approach is illustrative to the above stated). All this to highlight that the previous statement should be re-considered by authors.

3. It is of noticeable importance the fact that the group “others” was leaved out of the analysis since, as pointed out here, this category includes livestock feed, with expected high GHG emissions associated in the case of (e.g.) soybeans (due to LUC, not included in this study).

Using the same coefficients for a group of items (as authors did here for “others”) is a well-documented issue in IO studies (e.g. Steen-Olsen et al. 2014). This issue should be further discussed in this manuscript as it could greatly affect results. It is not enough with a general recommendation as the following one (lines 280-284):

“We therefore recommend that future research should build on the existing FAOSTAT emission datasets to include more specific emission data for these ‘other’ items, as many of these items are economically important in several major agricultural exporters (e.g. palm oil from Indonesia and Malaysia).”

4. Line 269-271: “In addition, the FAOSTAT is one of the most comprehensive datasets available and therefore best suited for our study’s purposes in terms of coverage, despite the availability of other input-output data that could address the issue of re-exporters.”

I fully agree about the quality and possitive attributes of FAOSTAT database and I understand that authors decided not using “other input-output data”, that can be useful for avoiding the re-exporters issue, because the resolution of FAOSTAT database (line 271). However, I have a question for authors here: Are you fully sure that the FAOSTAT trade matrix cannot be used for re-exports calculations?

Authors stated “To estimate emissions of the traded items, we used emission intensities provided by the FAOSTAT” but, apart of a lack of references in the text body to this source of information, when you go to the reference list (ref. nº 38), the link takes the reader to a pdf file with no quantitative information/data. It is very important for the consideration of this study to perfectly clarify the source of information used as well as a clear access to it.

5. Finally, I do miss a proper statistical analysis of the results. Even a sensitivity analysis could work. Please, consider this point.

Other considerations:

Introduction:

- Line 26: “emissions need to be drastically reduced at every stage of the food system from production to consumption.” I would also add post-consumption emissions here (e.g. waste management).
- Line 37: “Thus, this study aims to explore, for all food items at the global”. Please, refer to the supplementary information.

Results:

This section is, in general, well-structured and organized although there are few potential improvements in Figures and captions (see below):

- Line 60.61: “... due to the rise in agricultural productivity and fall in emission intensity of food

production in recent decades at the global scale (Figure S3). 12". What about population growth?

- Figure 1: Use bold letters for units, that would make easier the understanding for readers. Although authors refer to country codes UN ISO, I might suggest including a clarification, in the figure caption, for the few countries specifically included in the bar graphs. There are some easy to infer (e.g. CHN, USA...) but there are others difficult to get them (e.g. FLK, SSD..). Authors made something similar in Figure 3.
- Line 93: "This shows that per capita CBE is not always directly related to economic development and affluence as is sometimes implied for consumption in general 3" Maybe better "...and affluence as it is sometimes"
- Lines 98-99: "due to their larger populations, particularly for countries such as mainland China, India and Japan." This also applies for global regions shown above, right?
- Lines 120-121: "This trend in Brazil demonstrates the rapid growth in export-driven production emissions in the country with increasing trade 20".

Maybe worthy to mention the type of products exported from Brazil and how they are shaping these reported high emissions? One may ask here, why emissions from Brazil are so high If LUC derived GHG emissions are not included in this research?

- Line 164: "However, it was only second to African intra-regional imports in terms of emissions due to the higher average emission intensity of paddy rice in Africa". Please, Revise this construction.
- Line 197: "Thus, this implies that, were countries in Africa to increase productivity and close their yield gaps to meet their own food demands (assuming, however, that emission intensity remains constant) 11, emissions would be higher than if these countries were to continue relying on international trade." This is a rather long sentence difficult to understand. Please, revise.

Discussion

- First sentence: "In this study, we estimated for the first-time consumption-based emissions for the agriculture sector and emissions embodied in food trade for all countries over a span of three decades."

Is this fully right? Are there other (similar) previous research as those referred in the introduction? Please, be more precise pointing out the novelty of your research (e.g. temporal resolution, spatial one, number of agricultural items?). Because, as you later said ("Additionally, our study has produced several interesting findings that add new insights to on-going discussions over emissions accounting and responsibilities, specifically from the perspective of food systems"), there are other previous studies to refer to.

- Line 215-217: "Firstly, our findings challenge the general notion that more economically advanced countries tend to have higher levels of emissions embodied in per capita consumption and imports 3,7,23".

These results have to be taken with care as high C footprint agricultural products and processes (e.g.

LUC) were not included in the estimations.

- Line 240: “Thus, by extension, our study supports and justifies the growing call for food consumption to shift away from such products towards less emission-intensive food items in order to achieve climate mitigation targets more effectively 15,25–28.”

Maybe so large sentence to say: ... the growing call to consume less meat and dairy products, mainly in high income economies, to achieve climate mitigation targets more effectively.

- Line 243: “Lastly, our ‘local production’ scenario offers an interesting insight into the potential consequences to emissions if countries were to adopt policies that close their yield gaps and increase self-sufficiency 11”. Please, Check the grammar.

- After reading the following two statements:

Lines 261-263: “However, such simplifications are necessary in order for us to systematically assign emissions to re-exporters based on what is officially reported to the FAO, without the need to adjust and modify production and trade data as was done in other studies 30,31. “

and

Lines 359-316 (methods section): “While some studies have attempted to rectify this via adjustments or balancing of trade and production numbers, these attempts have mostly been done for specific food items and trading partners 30,31.”

I would encourage authors to carefully read the work by Bruckner et al. (2018) and reconsider the previous two statements afterwards.

Methods:

Please, see also all previous (major) comments stated above.

- Line 347: “Secondly, for any given country, we assumed that production for domestic consumption has the same characteristics as production for exports.”

This is reasonable but leads again to the re-exporters issue. Have authors considered the possibility of assuming footprint of exports as the weighted average of the footprint of production and imports together?

- Line 361-364: “As our study aims to look at the full spectrum of food items at the global scale, our approach was necessary in order for us to systematically assign trade flows (and thus emissions) to all countries despite this data gap.”

I would not say that this is “necessary” as there are other solutions to assign the trade flows. A key issue here is that authors, not only did not consider re-exports, as mentioned many times before, but, as far as I see, they did not assign the GHG emissions of feed to the specific livestock type eating that feed. This could produce important effects on the final assignment of emissions.

- Line 398: “This is because yield improvements through agricultural intensification can actually

result in a reduction in emission intensities per unit of product, provided that cropland expansion is avoided 42.”

Maybe also the use of inputs is also increased?

Overall, although the manuscript deals with one of the major concerns in agriculture and GHG research nowadays: the need to incorporate consumption-based estimations when calculating the GHG implications of national food systems, and this is a highly interesting and scientifically sound topic, methodological concerns exposed above, takes me to encourage authors for major revision prior re-considering again the potential of your work for publication in Nature Communications.

Yours sincerely,

References of this revision:

Bruckner, M., Wood, R., Moran, D., Kuschnig, N., Maus, V., Börner, J., 2019. FABIO — The Construction of the Food and Agriculture Biomass Input – Output Model. *Environ. Sci. Technol.* 53, 11302–11312. <https://doi.org/10.1021/acs.est.9b03554>

Cadarso, M.Á., Monsalve, F., Arce, G., 2018. Emissions burden shifting in global value chains – winners and losers under multi-regional versus bilateral accounting. *Econ. Syst. Res.* 30, 439–461. <https://doi.org/10.1080/09535314.2018.1431768>

Lenzen, M., Murray, J., Sack, F., Wiedmann, T., 2007. Shared producer and consumer responsibility - Theory and practice. *Ecol. Econ.* 61, 27–42. <https://doi.org/10.1016/j.ecolecon.2006.05.018>
Peters, G.P., 2008. From production-based to consumption-based national emission inventories. *Ecol. Econ.* 65, 13–23. <https://doi.org/10.1016/j.ecolecon.2007.10.014>

Steen-Olsen, K., Owen, A., Hertwich, E.G., Lenzen, M., 2014. Effects of Sector Aggregation on Co2 Multipliers in Multiregional Input–Output Analyses. *Econ. Syst. Res.* 26, 284–302. <https://doi.org/10.1080/09535314.2014.934325>

Su, B., Ang, B.W., 2011. Multi-region input–output analysis of CO2 emissions embodied in trade: The feedback effects. *Ecol. Econ.* 71, 42–53. <https://doi.org/10.1016/j.ecolecon.2011.08.024>

Piñero, P., Pérez-Neira, D., Infante-Amate, J., Chas-Amil, M.L., Doldán-García, X.R., 2020. Unequal raw material exchange between and within countries: Galicia (NW Spain) as a core-periphery economy. *Ecol. Econ.* 172, 106621. <https://doi.org/https://doi.org/10.1016/j.ecolecon.2020.106621>

Reviewer #2 (Remarks to the Author):

Consumption-Based Greenhouse Gas Emissions of the Agriculture Sector

(Manuscript ID: NCOMMS-20-24771-T)

This manuscript contributes to the literature by examining and applying the consumption-based approach in measuring the amount of Greenhouse Gas (GHG) emissions of the agricultural sector. The consumption-based approach is, generally, less commonly employed in the literature compared to the more conventional production-based approach. I enjoyed reading this manuscript, and I can see a decent contribution to the literature, which would allow analysts and policy-makers to better comprehend the patterns and characteristics of GHG emissions, and to develop policies to reduce GHG emissions through an international system characterized by relatively closer production and consumption locations. I have the following comments and suggestions.

International trade in agricultural and food products are determined by several factors, including technological and factor-based comparative advantage, market size, development level, and several bilateral factors that reduce trade flows (e.g., trade costs, including tariff and non-tariff barriers) or promote trade flows (e.g., preferential trade agreements, colonial ties, linguistic ties). In the empirical economic literature, several studies highlighted the significance of these factors in explaining the trade patterns in agricultural and processed food products using the gravity model (Olper and Raimondi, 2008; Sun and Reed, 2010; Ghazalian, 2017). It would be beneficial to understand to what extent the consumption-based approach would benefit from incorporating such factors into the analysis, and to examine their corresponding relevance to policies aiming at reducing GHG emissions.

Cristea et al. (2013) show that the inclusion of transportation into the analysis would significantly change the ranking of countries by emissions per dollar of trade. Can the author(s) make a correspondence through their analysis, and further explain how transportation is accounted for and how it affects the results?

The “local scenario” could have been made more realistic by incorporating more economic factors in the simulations, and it could have benefited from further discussions of policies that would make locations of production and consumption of agricultural products closer to each other. There are some trade-offs between reducing GHG emissions and losses in economic efficiency. These losses could stem from economic situations that are not consistent with the prevalence of comparative advantage, where comparative advantage implies that one country produces a good or service for a lower opportunity cost than other countries.

The actual trends in international trade are largely determined by economic, geo-economic and political forces, and altering these patterns should be discussed in the context of reducing GHG emissions. Also, this study would benefit from a brief discussion of “optimal” outcomes that jointly encompass the economic factors (or economic welfare) and the objective of reducing GHG emissions.

Some details about the local scenario are required. For instance, it is indicated that, through this scenario, there are reductions in imports and increases in local production to meet consumption needs. It is not clear how the latter is determined through the analysis. Also, it is noted that a country increases local production and decreases imports if it was already producing this commodity. Is there a certain production threshold for “already producing” or any minimal production amount is considered?

The second assumption is not clear, and its association with the note - that many countries still face several environmental and socio-economic barriers towards increasing productivity and achieving potential crop yields - needs further clarification. Also, I wonder if it is possible to somehow proxy such barriers in the simulation model.

Kander et al. (2015) propose an improvement to the consumption-based carbon accounting that takes technology differences in exporting sectors into consideration. Do you expect that this inclusion would impact your results and conclusion?

Finally, I wonder about the role of (horizontal) Foreign Direct Investment (FDI) in bringing production and consumption locations closer to each other in some cases, as analyzed through the proximity-concentration trade-off framework (Brainard, 1997; Ghazalian and Furtan, 2009). International trade can be substituted in this case by local production of foreign affiliates of multinational enterprises. The author(s) could add a corresponding note in this context.

References

Brainard, S.L., 1997. An Empirical Assessment of the Proximity-Concentration Trade-off between Multinational Sales and Trade. *American Economic Review*, 87(4), pp.520-544.

Cristea, A., Hummels, D., Puzello, L. and Avetisyan, M., 2013. Trade and the Greenhouse Gas Emissions from International Freight Transport. *Journal of Environmental Economics and Management*, 65(1), pp.153-173.

Ghazalian, P.L., 2017. The Effects of NAFTA/CUSFTA on Agricultural Trade Flows: An Empirical Investigation. *Canadian Journal of Agricultural Economics*, 65(2), pp.219-248.

Ghazalian, P.L. and Furtan, W.H., 2009. CUSFTA Effects: A Joint Consideration of Trade and Multinational Activities. *The Journal of International Trade & Economic Development*, 18(4), pp.487-504.

Kander, A., Jiborn, M., Moran, D.D. and Wiedmann, T.O., 2015. National Greenhouse-Gas Accounting for Effective Climate Policy on International Trade. *Nature Climate Change*, 5(5), pp.431-435.

Olper, A. and Raimondi, V., 2008. Agricultural Market Integration in the OECD: A Gravity-Border Effect Approach. *Food Policy*, 33(2), pp.165-175.

Sun, L. and Reed, M.R., 2010. Impacts of Free Trade Agreements on Agricultural Trade Creation and Trade Diversion. *American Journal of Agricultural Economics*, 92(5), pp.1351-1363.

Reviewer #3 (Remarks to the Author):

“Consumption-based greenhouse gas emissions of the agriculture sector”

- What are the noteworthy results?

This manuscript provides consumption-based, national-level estimates of greenhouse gas emissions from the farming stage of human-food items. Consumption is evaluated as production plus imports minus exports. These three food quantities, as well as the emission intensities of the food products, are all taken from the public FAOSTAT website, a global, national-level database.

The authors provide the results of this calculation for the years 1987 to 2015.

The authors also compare these 'observed' consumption-based emissions with those in a hypothetical 'local production' scenario.

- Does the work support the conclusions and claims, or is additional evidence needed?
- Are there any flaws in the data analysis, interpretation and conclusions? - Do these prohibit publication or require revision?

There is no obvious technical flaw I can find.

The 'local production' scenario is not meant to be realistic as the only constraint applied is that the country produces the crop already - for example, land availability, costs and other resources availability are not considered. So this is the same approach taken when calculated 'land savings' or 'water savings'. This is useful but has recognised and important limitations, such as its lack of practical meaning.

- Is the methodology sound? Does the work meet the expected standards in your field?

The methods overall are sound but quite simplistic relative to other studies in this field. Indeed, previous studies have implemented more sophisticated methods, which e.g. correct for most of re-exports, use product-specific emission intensities (rather than categories), design realistic counterfactual scenarios, among other advantages.

- Is there enough detail provided in the methods for the work to be reproduced?

Yes.

- Will the work be of significance to the field and related fields? How does it compare to the established literature? If the work is not original, please provide relevant references.

In my view, this manuscript is not innovative or significant enough to warrant a publication in Nature Communications.

The concepts, approaches and assumptions are not new. Indeed many studies that are cited here in the references refer to similar work: Davis and Caldeira (ref.3) and Caro et al. (ref.4) on the importance of consumption-based accounting of GHG emissions; Caro et al. (ref. 25) on emissions embodied in meat trade; Watts et al. 2019 (not cited; reference: The 2019 report of The Lancet Countdown on health and climate change, The Lancet, 394, 10211, 2019, [https://doi.org/10.1016/S0140-6736\(19\)32596-6](https://doi.org/10.1016/S0140-6736(19)32596-6).) also provides the production-based GHG emissions of food products over time by country, but using a process-based model instead of the FAOSTAT data for emission intensity estimates.

Besides, the approach in terms of data collection/combination is also simple because all the data used in this manuscript comes from the same source; this has the advantage of (hopefully) inherent consistency but means that besides the calculation, there is no value added in terms of combining different datasets.

Overall, although it is interesting and an important issue, I believe the presented work is too incremental.

I suggest the authors significantly change the design of the study to make a more substantial addition to the literature.

Rebuttal letter

Reviewer #1

Dear authors,

Please, see below my reviewer report on your manuscript "Consumption-based greenhouse gas emissions of the agriculture sector":

The manuscript by Foong, Pradhan and co-authors has been now revised for potential publication in Nature Communications.

The study is, mainly due to the topic addressed, timely and of great interest for a broad audience. Authors assessed the demand-side GHG emissions in a global context, with a disaggregation of information in terms of regions and food items. A global assessment on demand-side emissions associated to food consumption worldwide is undoubtedly attractive and of great scientific and social interest.

One of the main outcomes of this research is the insight on the importance of being an agricultural-product exporter vs an importer country over the weight of consumption vs production GHG related emissions on the countries' GHG share. This is well exemplified by authors with a scenario approach.

The work is fully based on FAO statistics which, in principle, should make results robust and reliable. However, there are some major methodological constraints I would like to point out as they could prevent the publication of this manuscript in its current form. Some of these limitations were even recognized by authors.

We thank the reviewer for highlighting our manuscript as being timely and of great interest to a wide audience. We also highly appreciate the reviewer's detailed list of constructive and encouraging comments, all of which have now been addressed in the revised version.

1. Apparently, authors did not include land use change related GHG emissions in their estimations. I see this as a substantial methodological limitation that might impact on the final figures of GHG emissions, of both production and consumption. This is mentioned by authors in the following statement:

"Finally, the findings of our study are based on emissions data for agricultural production only, and we did not account for other stages of food systems such as land use change and post-production processes due to the lack of comprehensive data covering these other emission sources in the FAOSTAT databases."

As said, this is highly important and could substantially change their findings for (e.g.) consumption emissions of imports from Latin America (such as EU).

We acknowledge that land use change (LUC) is indeed a key component to food system emissions for several food items, with some well-known examples including beef and soy beans from Latin America.

However, linking LUC and deforestation-related emissions to specific agricultural items may introduce more uncertainties rather than improve the robustness of our findings on emissions. This is because of the complex interaction between different land uses, and the difficulty in establishing a cause and effect mechanism for all food items in a consistent manner, as acknowledged by several studies:

- Schierhorn et al. (2016). The dynamics of beef trade between Brazil and Russia and their environmental implications.
 - "Attribution of CO2 emissions from deforestation to agricultural goods is complicated by the complex interactions among different land uses."
- Moran and Wood (2014). Convergence between the EORA, WIOD, EXIOBASE, and OPENEU's consumption-based carbon accounts.
 - "...land-use change and forestry emissions are generally not included due to the difficulty in establishing cause and effect mechanisms in an MRIO framework."

While some studies (e.g., Sandstrom et al. (2018), who also studied agricultural GHG emissions related to food), have attempted to address this very issue of LUC emissions, they do so by broadly linking deforestation emissions to cropland and pasture area. These links, however, are not commodity-specific, which again raises the question of how these different land use types can be linked to particular items in the absence of such information.

Another study by Hawkins et al. (2016), which utilised FAOSTAT emissions data to estimate carbon emission factors related to crop and livestock products, also excluded LUC-related emissions due to the uncertainties and extensive assumptions necessary to calculate and include such emissions.

Additionally, we did a screening of other major databases and input-output tables that contain both agricultural trade and emissions data to see if these sources might contain LUC-related emissions. These sources include EXIOBASE, Eora, WIOD, and GTAP. We found that, to the best of our understanding, none of these contain such emissions, and at the temporal and spatial scale (i.e., global, and over a continuous period of three decades) of product-specificity that we seek for. This was further confirmed via a discussion we had with an expert on MRIO analyses, Dr. Martin Bruckner from the Vienna University of Economics and Business. He confirmed that such data is indeed unavailable, particularly for EXIOBASE, a dataset which would have suited our study design the closest in terms of product-specificity. He did, however, point out that a new version of the FABIO input-output model covering all sources of GHG emissions in agricultural production is being developed, which could be of interest for future research into LUC-related agricultural emissions.

Nevertheless, in our revised manuscript, we attempted to address this gap by conducting a simple test on how agricultural land use emissions, specifically emissions from forestry and other land use (FOLU), would change when adjusting for trade flows. The findings of this test are presented in Figure 5 and reported in our revised manuscript as follows:

“In addition to our main focus on farm-gate emissions, we also tested how agricultural land use emissions, using FAOSTAT land use emissions data ³², would change when adopting a trade-adjustment approach for the year 2015 (see Methods).

We found that several major agricultural exporters had relatively high agricultural land use emissions as well as remarkable decreases in such emissions when adjusted for trade flows (Figure 5; see also Supplementary Fig. 10 for per capita results). Indonesia’s agricultural land use emissions, for example, decreased by 471.9 Mt CO_{2e}/yr when adjusted for trade flows, representing more than 40% of its original agricultural land use emissions. In Brazil, this decrease after adjusting for trade flows was 120.1 Mt CO_{2e}/yr, which was more than 20% of its original agricultural land use emissions.

Conversely, some countries with relatively low agricultural land use emissions saw a considerable increase in emissions when adjusted for trade flows (Figure 5; Supplementary Fig. 10). For example, the agricultural land use emissions of mainland China and India were only 2.51 Mt CO_{2e}/yr and 9.32 Mt CO_{2e}/yr respectively, but these figures increased by a factor of more than 50 and 10 respectively when trade flows were included. We also saw similar results for a number of import-dependent countries, such as several European countries (e.g., Italy and Spain, where the increase was more than 60 times, the highest increase among all countries in Europe), and small island states (e.g., Singapore, where agricultural land use emission was less than 0.1 Mt CO_{2e}/yr, but the increase was more than 100 times with trade adjustments). These results reflect the export- and import-driven nature of land use emissions in many countries, particularly where agricultural land use emissions are high, such as Brazil and Indonesia, or where there is a high dependency on agricultural imports, such as mainland China and Europe.”

Furthermore, we added a brief discussion on the limitations of incorporating emissions related to pre-production processes such as FOLU and land use change as follows:

“Furthermore, our study is mostly focused on emissions from the agricultural production stage, and does not consider emissions from other pre- and post-production processes. One notable process missing from our main TAE estimates is emissions from FOLU and land use change, which are key contributors to food system emissions alongside production stage processes

^{5,39,40}. We attempted to address this gap by testing how agricultural emissions from FOLU would change when adjusted for trade flows. To do this, we assumed that all food items drive land use emissions proportionally to their production volumes due to the recognised limitations in linking FOLU emissions to specific agricultural items. Specifically, this is because of the difficulties in establishing a cause-and-effect mechanism between food items and land use change in a consistent manner ^{22,41,42}.”

Emissions from pre-production processes (e.g. inputs production) and transport are not included in the analysis and this is a methodological disadvantage comparing with “other input-output data” authors referred in line 271.

Many thanks for the comment. We acknowledge that our method focuses on emissions generated within the farm gate. However, we do not consider it as a disadvantage in comparison with “other input-output data”, because we are primarily interested in *trade-adjusted* agricultural emissions. Therefore, we decided to restrict our methodology to only one stage, i.e., the agricultural production stage, which constitutes the largest share of emissions from the entire agricultural system (Poore and Nemecek, 2018). However, we also included a simple test on agricultural land use emissions in the revised version. Please see our previous comment.

2. The following statement (lines 263-268): “We further postulate that tracing the precise pathways of exported items may leave re-exporters out of the emissions accounting picture altogether. From an ethical point of view, we believe that such re-exporters should also bear some responsibility to emissions embodied in global trade and consumption, as they ultimately benefit from such trade economically despite not consuming the items themselves.” Took me to the following considerations:

We thank the reviewer very much for the insightful comments on this topic. We addressed all the comments in our revised manuscript. Mainly, we avoid discussion on re-export issues and account for the considerations mentioned below.

- Bilateral trade input-output (point 2 of Methods section) has a very specific meaning, which comes from a still open discussion regarding to both methodological and policy-analysis aspects. This should be mentioned somehow in the manuscript. On the contrary, statements such as the above mentioned could be understood as a way to somehow avoid recognition to the Multi-Regional Input-Output (MRIO) allocation approach, as well as other approaches that deal with this issue (e.g. “shared responsibility”, see Lenzen et al. 2007).

As suggested, we mentioned the methodological and policy-analysis aspects of the bilateral trade input-output (BTIO) approach in the revised manuscript. Additionally, we acknowledge the importance of the Multi-Regional Input-Output (MRIO) approach in capturing the concept of a ‘consumption-based analysis’ (CBA) more accurately. In the revised manuscript, we avoid discussion on responsibility issues. These changes are reflected in the revised manuscript as follows:

“Interpretation of our findings also warrants some understanding of the different ways in which emissions embodied in the consumption of food and agricultural products can be estimated. These include: (1) the multi-regional input-output (MRIO) approach that explores trade for final consumption, and (2) the BTIO approach, which we applied in this study ^{46,47}. Although the MRIO approach more accurately captures the concept of a ‘consumption-based analysis’, the BTIO approach is more transparent and suited for analysing bilateral political agreements as well as trade and climate policies ¹². This is because it correlates directly with monetary bilateral trade data, and considers bilateral trade without splitting it into intermediate and final consumption ¹². Furthermore, the BTIO approach is also preferable for assessing border tax adjustments ⁴⁸, and is increasingly being used to test the pollution haven hypothesis ⁴⁶.”

- If authors address emissions from consumption, they should stick to that. If authors want to attribute impacts and responsibilities along the value chain, they should carry out a specific analysis of that, studying the share of emissions and the economic value that corresponds to each stage of the chain.

Many thanks for the comments. We would like to keep our study’s focus on consumption, rather than responsibilities along the value chain. Thus, we drop our statements on “responsibilities” throughout our

revised manuscript, and have even offered a clarification of the type of “consumption” that we refer to, i.e., consumption based on the BTIO approach. Please also see above response.

- The concept of “Consumption-based approach” (CBA), pivotal for this research (appearing even in the title), is of great interest. As pointed out by authors, it (undoubtedly) should be generalized in the field of GHG mitigation within the agro-food sector. However, when authors talk about CBA, the focus should be on final consumption so re-exporters would be less important. Of course, re-exporter and intermediates have their responsibility in GHG emissions (and mitigation) but there are appropriate indicators to do so and “bilateral trade input-output” could be a methodological option to address this issue.

The CBA concept and approach has been explored before by several authors. Peters et al. (2008) already stated its importance due to potential “carbon leakage”. In the input-output (IO) community, researchers firstly worked with available national input-output tables. Then, authors utilized global Multi-Regional Input-Output (MRIO) modelling tools (e.g. Eora, Exiobase, Fabio, etc.). The different tools and approaches used led to different notions of consumers’ footprints, depending on the IO models used and how bilateral trade is conceived and used. In this context, Kanemoto et al. (2012), Su & Ang. (2011) and Cadarso et al. (2018) worked deeply on these different perspectives. These authors distinguished two approaches: bilateral trade input-output (BTIO) and MRIO. This basically means that there are two ways of calculating the CBA (or consumers’ footprint) from an IO perspective. Each of them based on different allocation procedures for products in an international trade. As said, this discussion is still open and beyond methodological aspects (e.g. what to do with re-exporters), both approaches are useful. The advantage of MRIO is that it is closer of CBA as emissions are assigned to a final consumer (no matter of the number of borders crossed by a certain good). Piñero et al. (2020) considered all this but focusing on raw materials instead of emissions (but their approach is illustrative to the above stated). All this to highlight that the previous statement should be re-considered by authors.

We thank the reviewer very much for the insightful comments on the differences between the BTIO and MRIO approaches, and how choosing either one of these would affect our interpretation of the CBA concept. The reviewer’s suggestions on references have also been very helpful in shedding light to this concept.

We have now made significant adjustments to the text (examples of which we have alluded to in the previous comments), and even to the title to clarify our study’s purpose. In particular, with reference to Kanemoto et al. (2012), we have decided that the ‘trade-adjusted emission inventories’ concept fits more closely with our methodology than our initial objective of a CBA (i.e., looking at ‘final consumption’). This is because our methodology treats bilateral trade flows exogenously, i.e., we assign exports and imports (including intermediate products) directly to all trading partners, irrespective of where the final consumption actually takes place (Piñero et al., 2020). Conversely, a CBA would be more suited with a MRIO approach, which was not employed in our study.

That being said, we do not rule out our approach as being ‘consumption’-focused. Rather, as pointed out by Cadarso et al (2018), our BTIO approach is simply a different form of CBA or “consumer responsibility.”

3. It is of noticeable importance the fact that the group “others” was leaved out of the analysis since, as pointed out here, this category includes livestock feed, with expected high GHG emissions associated in the case of (e.g.) soybeans (due to LUC, not included in this study).

We would like to point out that we did not exclude the “other” food group from our analysis, and have indeed included these items under the group name “others.” This is because the FAOSTAT only provides emissions data for a limited number of food groups (14 in total), and not for these ‘other’ agricultural items. We do acknowledge, however, that this was not spelled out clearly in our methodology section, and have now made the necessary adjustments as follows:

“We added another food group called ‘others’ to include items not originally covered by FAOSTAT emission groupings (e.g., vegetables, fruits, and oilcrops). To calculate the emission intensities of ‘others’, we utilised FAOSTAT’s total agricultural emissions data⁶⁴, and proceeded with the following method: we first calculated the total emissions from ‘others’ by subtracting

total agricultural emissions by the emissions from the 14 groups for each country and year. Afterwards, we obtained the emission intensity of 'others' by dividing the calculated emissions with the production volumes of these items. We also estimated the emission intensity at the regional and global levels by repeating the above-mentioned steps."

Regarding LUC, please see our response to the first comment.

Using the same coefficients for a group of items (as authors did here for "others") is a well-documented issue in IO studies (e.g. Steen-Olsen et al. 2014). This issue should be further discussed in this manuscript as it could greatly affect results. It is not enough with a general recommendation as the following one (lines 280-284):

"We therefore recommend that future research should build on the existing FAOSTAT emission datasets to include more specific emission data for these 'other' items, as many of these items are economically important in several major agricultural exporters (e.g. palm oil from Indonesia and Malaysia)."

We thank the reviewer very much for this comment, and for recommending this insightful paper by Steen-Olsen et al. (2014). According to their paper, aggregate CO₂ product multipliers (which is synonymous to our study's "emission intensity" for the "others" food group) can vary considerably depending on the level of sector (or food group) detail used to estimate the multiplier. However, this variability is less important for national-level total carbon footprint analyses such as ours, as the results of such analyses are "determined not from multipliers alone, but as the product of multipliers and consumption volumes." Steen-Olsen et al. (2014) therefore conclude that, for such analyses, "a higher level of detail will have some, but probably more limited, influence."

In light of this, we conclude that our method of aggregating "others" using the same coefficient is sufficiently sound, and that going into further detail may not yield much added value. We discuss this issue in our revised manuscript as follows:

"Although emission intensities can vary considerably depending on the level of detail used to estimate them, this variability is less of an issue for our analysis in which country-level emissions are estimated by both intensities and volumes³⁸. Furthermore, since these 'other' items are responsible for only 14.3% of total agricultural production emissions from 1986 to 2017, our study has already captured a majority of emissions based on food group-specific emission intensities."

4. Line 269-271: "In addition, the FAOSTAT is one of the most comprehensive datasets available and therefore best suited for our study's purposes in terms of coverage, despite the availability of other input-output data that could address the issue of re-exporters."

I fully agree about the quality and positive attributes of FAOSTAT database and I understand that authors decided not using "other input-output data", that can be useful for avoiding the re-exporters issue, because the resolution of FAOSTAT database (line 271). However, I have a question for authors here: Are you fully sure that the FAOSTAT trade matrix cannot be used for re-exports calculations?

We admit that the FAOSTAT trade matrix can be used for re-export calculations. This has been done in a paper by Kastner et al. (2011), who used a matrix algebra approach to estimate land and water footprints of a single country's (Austria's) consumption of soybeans.

Other papers have also used this approach in analysing trade flows and greenhouse gas emissions related to food systems, although these were limited to specific countries (e.g., de Ruiter et al. 2016 for the United Kingdom), regions (e.g., Sandstrom et al. 2018 for the European Union; Schierhorn et al. 2019 for the former Soviet Union), and commodities (e.g. Henders et al. 2015 for 4 items, namely beef, soybean, palm oil, and wood products). To the best of our knowledge, this approach has not been applied to the range of items, years, and geographical scales that our study aimed for (i.e. 153 primary agricultural items, 30 years, and for all countries).

For our study, we did not use the approach by Kastner et al. (2011) as we aimed to start with a simple and straightforward method to obtain a general overview of trade-adjusted agricultural emissions, using readily available data from FAOSTAT. However, considering the potential for this approach to be used

for our study's context, we acknowledge that future research could use this approach as a next step to further investigate the issue of re-exporters, and how this might affect our estimations of trade-adjusted agricultural emissions.

Authors stated "To estimate emissions of the traded items, we used emission intensities provided by the FAOSTAT" but, apart of a lack of references in the text body to this source of information, when you go to the reference list (ref. n° 38), the link takes the reader to a pdf file with no quantitative information/data. It is very important for the consideration of this study to perfectly clarify the source of information used as well as a clear access to it.

We did intend reference #38 to link directly to the PDF containing the background information of the FAOSTAT emissions data, but acknowledge that we should have also added a link directly to the data source itself. We have now made the appropriate changes as follows:

"To estimate emissions of the traded items, we used agricultural emission intensities provided by FAOSTAT ⁵⁸"

5. Finally, I do miss a proper statistical analysis of the results. Even a sensitivity analysis could work. Please, consider this point.

It is not clear to us what "a proper statistical analysis" would entail in our study's context. However, we agree that a sensitivity analysis was missing, and have now supplemented our revised manuscript with two sensitivity analyses that look at the use of different emission intensities.

In these sensitivity analyses, we explore how much our emission estimations would vary if we used different emission intensities. We included a new section on sensitivity analysis in the results section as follows:

"Our two sensitivity analyses reveal the robustness of our TAE estimates (Supplementary Figs. 7, 8, 9). In the first analysis, we used global emission intensities to calculate TAE where a non-producer country is concerned, instead of using regional emission intensities which was the case for the original analysis. By switching to global emission intensities, we thus assumed a more global, inter-connected food trade network where imports are not restricted to a country's region. We found that the switch to global emission intensities only produced a very small difference, with a mean increase in TAE of +0.2% from the original approach (Supplementary Fig. 7). We found similar results when looking at specific food groups (Supplementary Fig. 9). These results therefore confirm that our approach of using regional emission intensities was sufficient in capturing the wide-reaching nature of global food trade.

In the second analysis, we adopted a 'technology-adjusted' approach in calculating TAE to account for technological differences in carbon efficiencies in the exports of different countries ¹⁹. This approach has been proposed as an improvement to the conventional consumption-based accounting by more accurately reflecting how changes to national climate policies could affect overall global emissions ¹⁹. Similar to the first analysis, the difference in switching to this approach was minor, with a mean decrease in trade-adjusted emissions of -2.4% from the original approach (Supplementary Fig. 8). However, the technology-adjusted approach did produce some interesting patterns at the regional and food group-specific levels (Supplementary Figs. 8, 9). For example, Europe and North America had lower TAEs for ruminant meat under this approach, reflecting the higher efficiency of ruminant meat production in these regions ²⁰. We thus conclude that while our results were relatively stable even when incorporating a technology-adjusted approach, using such an approach also revealed several other noteworthy results, while at the same time reconfirming some of our earlier findings."

Figures for our sensitivity analyses results are provided in the supplementary information, specifically Supplementary Figs. 6-8.

Other considerations:

Introduction:

• Line 26: “emissions need to be drastically reduced at every stage of the food system from production to consumption.” I would also add post-consumption emissions here (e.g. waste management).

We have adjusted the text accordingly.

“Thus, to limit global warming well below 2°C as stipulated in the Paris Agreement, emissions need to be drastically reduced at every stage of the food system from pre-production to post-consumption³.”

• Line 37: “Thus, this study aims to explore, for all food items at the global”. Please, refer to the supplementary information.

We refer to the supplementary information accordingly.

“Thus, we aim to explore, for all food items at the global, regional and national levels (Supplementary Tables 1, 2), recent trends in (i) trade-adjusted agricultural emissions and how they differ from the production-based approach, and (ii) emissions embodied in trade flows between producer and consumer regions.”

Results:

This section is, in general, well-structured and organized although there are few potential improvements in Figures and captions (see below):

Many thanks for finding this section well-structured and organised. We further improve the section following the comments.

• Line 60.61: “... due to the rise in agricultural productivity and fall in emission intensity of food production in recent decades at the global scale (Figure S3). 12”. What about population growth?

Regarding the question of population growth, we would like to clarify that the increase in emissions in *total absolute terms* is indeed due to population growth, as a growing population entails a higher demand for food and agricultural production. This increase in agricultural production happens in two ways: agricultural expansion and agricultural intensification. Specifically, intensification means that for every unit of agricultural input and related emissions, the volume of output should increase. This increase in ‘efficiency’ would then result in a decrease in emission intensity, which we have observed (Supplementary Fig. 3).

Thus, the fall in *per capita emissions* as we noted in our results perhaps indicates that, at the global scale, the increase in agricultural production is mostly due to intensification rather than expansion. We have now included these clarifications into the revised version of the manuscript as follows:

“Global trade-adjusted agricultural emissions (TAE), in total absolute terms, increased from 3.87 Gt CO_{2e}/yr in 1987 to 5.02 Gt CO_{2e}/yr in 2015. This increase in emissions corresponds to the rise in the global agricultural production volumes driven by population growth and changing diets, resulting in an increase in production-based emissions (PBE) (Supplementary Figs. 1, 2). However, in per capita terms, we found a decrease in TAE from 0.77 t CO_{2e}/cap/yr in 1987 to 0.68 t CO_{2e}/cap/yr in 2015. This fall in per capita TAE is because of the increase in agricultural production that is largely driven by efficiency improvements afforded by agricultural intensification^{20,21}, thus resulting in a decrease in emission intensity (Supplementary Fig. 3).”

• Figure 1: Use bold letters for units, that would make easier the understanding for readers. Although authors refer to country codes UN ISO, I might suggest including a clarification, in the figure caption, for the few countries specifically included in the bar graphs. There are some easy to infer (e.g. CHN, USA...) but there are others difficult to get them (e.g. FLK, SSD...). Authors made something similar in Figure 3.

We thank the reviewer for pointing these out. We have now included a full clarification of all abbreviations used in Figure 1, along with other formatting adjustments. We updated our figure captions as follows:

“Figure 1: Trade-adjusted agricultural emissions (TAEs) in 1987 and 2015. Total absolute TAEs are shown on the left, and per capita TAEs on the right. A darker red colour indicates higher TAE levels, whereas dark grey colours indicate countries with no available data (e.g., Greenland). In addition, the bar graphs show the top five countries with the largest total absolute

and per capita TAEs, in decreasing order from left to right. The bars also consist of the relative contributions of each food group to these TAEs. Country names are shown according to their respective UN ISO 3166-1 alpha-3 codes (AUS=Australia; BRA=Brazil; BRB=Barbados; CAF=Central African Republic; CHN=China, mainland; FLK=Falkland Islands; IDN=Indonesia; IND=India; MNG=Mongolia; NZL=New Zealand; SSD=South Sudan; SUN=USSR; USA=United States of America; URY=Uruguay).”

- Line 93: “This shows that per capita CBE is not always directly related to economic development and affluence as is sometimes implied for consumption in general 3” Maybe better “...and affluence as it is sometimes”

Many thanks, we changed it accordingly.

- Lines 98-99: “due to their larger populations, particularly for countries such as mainland China, India and Japan.” This also applies for global regions shown above, right?

Yes, in addition to reflecting a regions’ consumption patterns, the low levels of per capita emissions in regions such as Asia are also a result of large population sizes and their relatively low food consumption per capita. The reverse is also true for regions such as Oceania that display relatively higher levels of per capita emissions, i.e., lower population density and higher consumption of more emission-intensive items. We highlighted this aspect in our revised manuscript as follows:

“It is also notable that most countries in Asia, despite having relatively large total absolute TAEs, generally had low levels of per capita TAE (<0.5 t CO_{2e}/cap/yr in 2015 on average) due to their larger populations and relatively low food consumption per capita, particularly for countries such as mainland China, India, and Japan.”

- Lines 120-121: “This trend in Brazil demonstrates the rapid growth in export-driven production emissions in the country with increasing trade 20”.

Maybe worthy to mention the type of products exported from Brazil and how they are shaping these reported high emissions? One may ask here, why emissions from Brazil are so high If LUC derived GHG emissions are not included in this research?

We fully agree with the reviewer’s suggestion on elaborating on Brazil’s exports and how these relate to the export emissions that we have observed. These high emissions are related to export of ruminant meat. Please find below a short summary regarding this issue, parts of which have been included in our revised manuscript:

“ Additionally, some major exporters saw a rising gap between PBE and TAE from 1987 to 2015. In Brazil, for example, PBE in total absolute terms was only 3.08 Mt CO_{2e}/yr higher than TAE in 1987, but this gap widened by 45.6 Mt CO_{2e}/yr in 2015. This trend in Brazil demonstrates the rapid growth in export-driven production emissions in the country with increasing trade ²⁸. For Brazil, the export emissions of ruminant meat tripled from 10.007 Mt CO_{2e}/yr in 1987 to 34.453 Mt CO_{2e}/yr in 2015. These export emissions were much larger than those of the second largest contributor to Brazilian export emissions, i.e., ‘others’, of which soybeans were the dominant items. Although soybean exports were larger than those of ruminant meat during both periods, the entire ‘others’ food group to which soybeans belong only contributed 1.229 Mt CO_{2e}/yr and 12.727 Mt CO_{2e}/yr in 1987 and 2015 respectively, because of a lower emission intensity than ruminant meat.”

Thus, our interpretation of Brazil’s “rapid growth in export-driven production emissions” do not come as a surprise as ruminant meat have the largest emission intensities compared to other food groups, due to the inclusion of processes such as enteric fermentation in FAOSTAT’s calculation of emission intensities. This is despite the fact that LUC was excluded from our analyses: in fact, the inclusion of LUC-related emissions would have increased the emission contributions of soybeans and other products in the ‘others’ food group, which would have given further support to our interpretation above in lines 120-121.

Please also see our previous response on LUC emissions.

- Line 164: “However, it was only second to African intra-regional imports in terms of emissions due to the higher average emission intensity of paddy rice in Africa”. Please, Revise this construction.

We have adjusted the text accordingly for clarity.

“However, in terms of emissions, Asia was only second to African intra-regional imports. Two reasons might explain this: (1) rice was the main contributor to Africa’s import emissions, and (2) Africa’s average emission intensity of paddy rice (1.14 kg CO_{2e}/kg product) was higher than that of Asia (0.90 kg CO_{2e}/kg product).”

- Line 197: “Thus, this implies that, were countries in Africa to increase productivity and close their yield gaps to meet their own food demands (assuming, however, that emission intensity remains constant) 11, emissions would be higher than if these countries were to continue relying on international trade.” This is a rather long sentence difficult to understand. Please, revise.

In the revised version this sentence is deleted because it refers to our ‘local production’ scenario. We have removed this scenario from our revised manuscript to address the comments from other reviewers.

Discussion

- First sentence: “In this study, we estimated for the first-time consumption-based emissions for the agriculture sector and emissions embodied in food trade for all countries over a span of three decades.”

Is this fully right? Are there other (similar) previous research as those referred in the introduction? Please, be more precise pointing out the novelty of your research (e.g. temporal resolution, spatial one, number of agricultural items?). Because, as you later said (“Additionally, our study has produced several interesting findings that add new insights to on-going discussions over emissions accounting and responsibilities, specifically from the perspective of food systems”), there are other previous studies to refer to.

Yes, we agree with the reviewer that there are indeed other studies that investigated the environmental footprints of food-related agricultural consumption, but to varying levels of detail. Thus, we have decided to rephrase the sentences mentioned by the reviewer to take into account these other studies. Specifically, we have now avoided mentioning our study as being a “first-time” attempt at measuring agricultural consumption-based emissions, and have instead opted to highlight the novelty and consistency of our study’s methodology, with reference to other similar studies. Our revised statement reads as follows:

“Our study adds new insights to on-going discussions over emissions accounting at the national level, with a focus on food and the agricultural sector. Specifically, we calculated emissions within the farm-gate that are embodied in agricultural trade flows, and used these to estimate TAE at the country level. To do this, we used a novel and consistent BTIO approach in which we explored all countries and all food items available from FAOSTAT. This approach extends previous studies that have also used a similar ‘trade-adjustment’ approach to estimate emissions, but these studies have so far limited their analyses to specific food items or geographical areas ^{16–18}.”

- Line 215-217: “Firstly, our findings challenge the general notion that more economically advanced countries tend to have higher levels of emissions embodied in per capita consumption and imports 3,7,23”.

These results have to be taken with care as high C footprint agricultural products and processes (e.g. LUC) were not included in the estimations.

Please see our responses above on emissions from other processes such as LUC. We have also adjusted the text to highlight the need for care when interpreting our results. Further, we restructure our statement to reflect that we only consider emissions within the farm-gate as follows:

“Additionally, our findings challenge the general notion that more economically advanced countries tend to have higher levels of emissions embodied in per capita consumption and imports ^{9,13,33}. Our results show that this is not always the case when considering emissions within the farm-gate, as several ‘low-income’ or ‘emerging and developing’ economies also display some of the highest per capita TAEs.”

• Line 240: “Thus, by extension, our study supports and justifies the growing call for food consumption to shift away from such products towards less emission-intensive food items in order to achieve climate mitigation targets more effectively 15,25–28.”

Maybe so large sentence to say: ... the growing call to consume less meat and dairy products, mainly in high income economies, to achieve climate mitigation targets more effectively.

We adjusted the text to reflect a need to consume healthy diets that consist of a low amount of meat and dairy products. This need to consume healthy diets is true for all countries. Thus, we did not explicitly mention high income countries in the revised version as follows:

“Concurrently, our results also clearly show the major contribution of animal-source food, particularly of ruminant meat and milk products, in overall emissions embodied in consumption and trade in general. Thus, by extension, our study supports and justifies the growing call to consume healthy diets consisting of a low amount of emission-intensive meat and dairy products, to achieve climate change mitigation targets more effectively ^{3,16,23,35–37}.”

• Line 243: “Lastly, our ‘local production’ scenario offers an interesting insight into the potential consequences to emissions if countries were to adopt policies that close their yield gaps and increase self-sufficiency 11”. Please, Check the grammar.

Many thanks. This comment refers to our ‘local production’ scenario, which we have removed from our revised manuscript.

• After reading the following two statements:

Lines 261-263: “However, such simplifications are necessary in order for us to systematically assign emissions to re-exporters based on what is officially reported to the FAO, without the need to adjust and modify production and trade data as was done in other studies 30,31. “

and

Lines 359-316 (methods section): “While some studies have attempted to rectify this via adjustments or balancing of trade and production numbers, these attempts have mostly been done for specific food items and trading partners 30,31.”

I would encourage authors to carefully read the work by Bruckner et al. (2018) and reconsider the previous two statements afterwards.

Having carefully re-read the work by Bruckner et al. (2019) in more detail, we agree that the above-mentioned lines need to be adapted, considering the similar time frames and assumptions used to handle import-export data inconsistencies. Accounting for this, we deleted those lines in the revised manuscript, and included the following line to recognise the work by Bruckner et al. (2019).

“Currently, only a limited number of studies have applied trade-adjusted approaches to estimate emissions from the agricultural sector ¹⁵, ...”

There are, however, two major differences between the FABIO model used by Bruckner et al. (2019) and our approach that we would like to point out:

1. The FABIO model is centred around FAOSTAT’s Commodity Balance Sheet (CBS), which we did not use in our paper. This is because of the difficulties that we found in linking the aggregation of commodities in the CBS to FAOSTAT’s more commodity-specific emissions data. For example, the CBS commodity group ‘Milk - Excluding Butter’ aggregates milk products from various livestock types together, whereas the FAOSTAT emissions dataset provides livestock-specific emission intensities of milk (e.g., ‘Milk, whole fresh cow’, ‘Milk, whole fresh goat’).

2. The FABIO model incorporates data from UN Comtrade to cover FAOSTAT's gap in bilateral trade data for fish and aquaculture. We did not do this to avoid (potentially) introducing more uncertainties into our analysis from the use of different datasets. Furthermore, the UN Comtrade data is only available in monetary units, and not in physical weight units which is necessary for us to be able to link both trade and emission intensities data.

Methods:

Please, see also all previous (major) comments stated above.

- Line 347: "Secondly, for any given country, we assumed that production for domestic consumption has the same characteristics as production for exports."

This is reasonable but leads again to the re-exporters issue. Have authors considered the possibility of assuming footprint of exports as the weighted average of the footprint of production and imports together?

We did not consider the use of weighted averages of production and import footprints, as this would violate one of the key assumptions of our paper, i.e., we assumed exports to come entirely from the exporter's production, or, if the exporter does not produce the item, entirely from the exporter's immediate region. In other words, we do not need to calculate weighted averages as we assume exports not to be a mix of both domestic production and imports.

Additionally, we have now reformulated the text for clarity as follows:

"Our BTIO approach is based on the following assumptions: for an exporter country, we assumed that the exports originated from the country's own domestic production. If the exporter country did not produce the export item (i.e., a non-producer country), we assumed that the exporter would have imported the item from another country within the same region. If the region did not produce the item, only then did we assume that the item was imported from elsewhere globally. These assumptions are important when considering the types of emission intensities to use to calculate emissions embodied in trade."

- Line 361-364: "As our study aims to look at the full spectrum of food items at the global scale, our approach was necessary in order for us to systematically assign trade flows (and thus emissions) to all countries despite this data gap."

I would not say that this is "necessary" as there are other solutions to assign the trade flows. A key issue here is that authors, not only did not consider re-exports, as mentioned many times before, but, as far as I see, they did not assign the GHG emissions of feed to the specific livestock type eating that feed. This could produce important effects on the final assignment of emissions.

We agree with the reviewer that there are other solutions (for e.g., the approach by Kastner et al. (2011), which we mentioned in a previous response), and have now omitted this statement in the revised version. Regarding emissions due to livestock feed, we have implicitly considered the emissions associated with feed in our estimations, as our trade-matrix includes crops whose by-products are used as livestock feed (e.g., cake of groundnuts, sesame, and soybeans; see also Supplementary Table 2).

Furthermore, the production of feed (and emissions associated with it) is considered a pre-production process (Vermeulen et al. 2012). As our study primarily focuses on the agricultural production stage (i.e., farm-gate stage), we have decided to exclude feed as a separate, unique process in estimating emissions related to livestock, similar to our exclusion of LUC-related emissions from our study's main focus.

- Line 398: "This is because yield improvements through agricultural intensification can actually result in a reduction in emission intensities per unit of product, provided that cropland expansion is avoided 42."

Maybe also the use of inputs is also increased?

This comment refers to our 'local production' scenario, which we have removed from our revised manuscript.

Overall, although the manuscript deals with one of the major concerns in agriculture and GHG research nowadays: the need to incorporate consumption-based estimations when calculating the GHG implications of national food systems, and this is a highly interesting and scientifically sound topic, methodological concerns exposed above, takes me to encourage authors for major revision prior re-considering again the potential of your work for publication in Nature Communications.

We thank the reviewer for providing such comprehensive and constructive comments to our manuscript. We did our best to address these comments, which we believe has substantially improved our manuscript. We highly appreciate you for spending your valuable time in reviewing our manuscript.

Yours sincerely,

References of this revision:

Bruckner, M., Wood, R., Moran, D., Kuschnig, N., Maus, V., Börner, J., 2019. FABIO — The Construction of the Food and Agriculture Biomass Input – Output Model. *Environ. Sci. Technol.* 53, 11302–11312. <https://doi.org/10.1021/acs.est.9b03554>

Cadarso, M.Á., Monsalve, F., Arce, G., 2018. Emissions burden shifting in global value chains – winners and losers under multi-regional versus bilateral accounting. *Econ. Syst. Res.* 30, 439–461. <https://doi.org/10.1080/09535314.2018.1431768>

Lenzen, M., Murray, J., Sack, F., Wiedmann, T., 2007. Shared producer and consumer responsibility - Theory and practice. *Ecol. Econ.* 61, 27–42. <https://doi.org/10.1016/j.ecolecon.2006.05.018>

Peters, G.P., 2008. From production-based to consumption-based national emission inventories. *Ecol. Econ.* 65, 13–23. <https://doi.org/10.1016/j.ecolecon.2007.10.014>

Steen-Olsen, K., Owen, A., Hertwich, E.G., Lenzen, M., 2014. Effects of Sector Aggregation on Co2 Multipliers in Multiregional Input–Output Analyses. *Econ. Syst. Res.* 26, 284–302. <https://doi.org/10.1080/09535314.2014.934325>

Su, B., Ang, B.W., 2011. Multi-region input–output analysis of CO2 emissions embodied in trade: The feedback effects. *Ecol. Econ.* 71, 42–53. <https://doi.org/10.1016/j.ecolecon.2011.08.024>

Piñero, P., Pérez-Neira, D., Infante-Amate, J., Chas-Amil, M.L., Doldán-García, X.R., 2020. Unequal raw material exchange between and within countries: Galicia (NW Spain) as a core-periphery economy. *Ecol. Econ.* 172, 106621. <https://doi.org/https://doi.org/10.1016/j.ecolecon.2020.106621>

We thank the reviewer very much for this helpful list of references. We have read each reference carefully and have taken them into consideration in our revision of the manuscript.

References

Bruckner et al. (2019). FABIO - The Construction of the Food and Agriculture Biomass Input-Output Model. *Environmental Science and Technology*, 53, 11302-11312.

Cadarso et al. (2018). Emissions burden shifting in global value chains—winners and losers under multi-regional versus bilateral accounting. *Economic Systems Research*, 30, 439-461.

De Ruiter et al. (2016). Global cropland and greenhouse gas impacts of UK food supply are increasingly located overseas. *Journal of The Royal Society Interface*, 13.

Hawkins et al. (2016). Apples to kangaroos: A framework for developing internationally comparable carbon emission factors for crop and livestock products. *Journal of Cleaner Production*, 139, 460-472.

Henders et al. (2015). Trading forests: land-use change and carbon emissions embodied in production and exports of forest-risk commodities. *Environmental Research Letters*, 10, 125012.

Kanemoto et al. (2012). Frameworks for comparing emissions associated with production, consumption, and international trade. *Environmental Science and Technology*, 46, 172-179.

- Kastner et al. (2011). Tracing distant environmental impacts of agricultural products from a consumer perspective. *Ecological Economics*, 70, 1032-1040.
- Moran and Wood (2014). Convergence between the Eora, WIOD, EXIOBASE, and OpenEU's consumption-based carbon accounts. *Economic Systems Research*, 26:3, 245-261.
- Pinero et al. (2020). Unequal raw material exchange between and within countries: Galicia (NW Spain) as a core-periphery economy. *Ecological Economics*, 172, 106621.
- Poore and Nemecek (2018). Reducing food's environmental impacts through producers and consumers. *Science*, 360, 987-992.
- Sandström et al. (2018). The role of trade in the greenhouse gas footprints of EU diets. *Global Food Security*, 19, 48-55.
- Schierhorn et al. (2016). The dynamics of beef trade between Brazil and Russia and their environmental implications. *Global Food Security*, 11, 84-92.
- Schierhorn et al. (2019). Large greenhouse gas savings due to changes in the post-Soviet food systems. *Environmental Research Letters*, 14, 65009.
- Steen-Olsen et al. (2014). Effects of Sector Aggregation on CO2 Multipliers in Multiregional Input-Output Analyses. *Economic Systems Research*, 26, 284-302.
- Vermeulen et al. (2012). Climate Change and Food Systems. *Annual Review of Environment and Resources*, 37, 195-222.

Reviewer #2

This manuscript contributes to the literature by examining and applying the consumption-based approach in measuring the amount of Greenhouse Gas (GHG) emissions of the agricultural sector. The consumption-based approach is, generally, less commonly employed in the literature compared to the more conventional production-based approach. I enjoyed reading this manuscript, and I can see a decent contribution to the literature, which would allow analysts and policy-makers to better comprehend the patterns and characteristics of GHG emissions, and to develop policies to reduce GHG emissions through an international system characterized by relatively closer production and consumption locations. I have the following comments and suggestions.

We thank the reviewer for the encouraging comments on how our manuscript would be a decent contribution to the research and policy making domains. We are also very pleased to know that the reviewer enjoyed the read. All comments and suggestions have now been considered and incorporated into the revised manuscript, where appropriate.

International trade in agricultural and food products are determined by several factors, including technological and factor-based comparative advantage, market size, development level, and several bilateral factors that reduce trade flows (e.g., trade costs, including tariff and non-tariff barriers) or promote trade flows (e.g., preferential trade agreements, colonial ties, linguistic ties). In the empirical economic literature, several studies highlighted the significance of these factors in explaining the trade patterns in agricultural and processed food products using the gravity model (Olper and Raimondi, 2008; Sun and Reed, 2010; Ghazalian, 2017). It would be beneficial to understand to what extent the consumption-based approach would benefit from incorporating such factors into the analysis, and to examine their corresponding relevance to policies aiming at reducing GHG emissions.

We thank the reviewer for these ideas and considerations. In general, we agree that these are important factors to consider in discussions pertaining to trade and policy decision-making. Specifically, regarding the reviewer's question on how our approach would benefit from taking such factors into account, we do not see in what way they would influence or change the analysis of trade-adjusted emissions *per se*, since the presented approach is purely an adjustment in emissions accounting to make the emissions embodied in trade of agriculture products visible for a comprehensive set of products on a global scale. We acknowledge the merits of the gravity-border model in its various specifications and the references cited by the reviewer in explaining (regional) trade patterns and analysing the complexity of trade. We rather see that GHG emissions are currently not sufficiently decisive for agricultural trade patterns since they represent no market relevant factors (e.g., comparative advantage in any sense). Until now there are no GHG emissions-based instruments implemented in the agricultural sector anywhere to a substantial extent. Consequently, current trade patterns in agricultural and food products do not depend on those. But this is likely to change in the near future: according to Fellmann et al. (2018), 121 countries intend to commit to emissions reductions in the agricultural sector.

This connects to the reviewer's comment to examining their corresponding relevance to policies aiming at reducing GHG emissions. Under the current intended policy to consider only production-based emissions, also in the agricultural and food sector, and strictly limit the allowed emissions in producing countries, studies like Fellmann et al. (2018) demonstrate that this would lead to suboptimal results because of carbon leakage to other countries where reduction commitments are less strict. They conclude that for the European Union a considerable fraction of the reduced production-based emissions of the agricultural and food sector are leaked to other regions of the world where in many cases emissions intensities are higher. Our trade-adjusted emissions accounting approach would be suited to work against this effect since it would give incentives to produce agricultural and food products in regions with low emission intensity if national (or regional) consumption-based emission reduction targets were to be fulfilled. In this sense we expect international trade policies and regulations to adjust in various ways to enable trade patterns that are more favourable for reaching those targets. These could be, always under the condition of being allowed in higher-level regulatory frameworks like the WTO, tariff and non-tariff barriers and the like. But to propose and discuss such policies is beyond the scope of the present paper. Taking up your comments and considerations (also referring to your

comment below regarding “optimal” policies and welfare) we have added the following two paragraphs in the discussion section (separate locations):

“With these characteristics of the BTIO approach, we believe that our study provides a foundation for further research that seek to investigate the links between agricultural emission inventories, trade patterns, and international climate policies. For example, because our method is consistent with bilateral trade data, our results could be used to examine the effects of trade agreements and tariffs on emissions embodied in trade and overall emission inventories, and vice versa. This application would be an extension of the findings of previous research that have looked at the trade creation and diversion effects as well as border effects of trade agreements, specifically for the agricultural sector ⁴⁹⁻⁵¹.”

“Considering TAEs in national emission inventories for the agricultural sector acknowledges demand as a driver for emissions. Under current climate change mitigation frameworks, national or regional commitments focus on production-based emissions, with demand playing only a minor role. As a result, this could lead to a displacement of emissions overseas. For example, stricter emission reduction targets for the European Union have been found to lead to a considerable displacement of saved emissions abroad, where agricultural emission intensities are higher ⁵⁵. Our analysis in calculating trade-adjusted emissions in the agricultural sector could serve as an important element towards constructing consumption-based national or regional emission reduction targets. In combination with trade policy regulations, such as border adjustment measures allowed by World Trade Organization regulations, such consumption-based targets would provide incentives to consider emissions intensities in agricultural production and trade, consequently leading to more cost-effective emission reductions overall.”

Cristea et al. (2013) show that the inclusion of transportation into the analysis would significantly change the ranking of countries by emissions per dollar of trade. Can the author(s) make a correspondence through their analysis, and further explain how transportation is accounted for and how it affects the results?

We did not include transportation emissions because such emissions account for a relatively small share of overall agricultural and food supply chain emissions, in comparison to emissions from the production stage, which generally constitutes the largest share (Niles et al., 2018; Poore and Nemecek, 2018). For example, in a review by Vermeulen et al. (2012) on global food chain GHG emissions for the period 2004-2008, transport (including storage and packaging) only accounted for less than 3% of total emissions.

It is worth noting that the share of transportation emissions varies across food types, with the smallest being found in ruminants and the largest in less emission-intensive items such as poultry and vegetables (Poore and Nemecek, 2018). However, the overall share of transportation emissions across the entire spectrum of food groups is still relatively small compared to the production stage, even at the national (e.g., Boehm et al., 2018 for U.S. diets) and regional levels (e.g., Sandström et al., 2018 for EU diets).

Furthermore, Cristea et al. (2011) point out that transportation emissions, in emissions per dollar, cover less than 3% of the overall emissions in the food and agricultural sectors. Instead, transportation emissions matter more in other sectors such as manufacturing, in which transportation emissions are considerably high, particularly for machinery exports.

Considering the above, we therefore do not think that adding transportation emissions into our analysis would greatly affect our results. However, in our revised manuscript, we did highlight the role of food transport distance reduction in lowering food transport emissions and enabling more responsible food production and consumption as follows:

“Similarly, our study did not account for transportation emissions. Although food transport reduction through the regionalisation of food systems is an important climate change mitigation option ⁴³, transport constitutes a very small share of overall food system emissions ^{5,6,39,44}. Therefore, excluding its emissions would not make much difference in our results. Nevertheless, bringing producers and consumers closer through regionalised food systems would lead to more

responsible food production and consumption ⁴⁵, thus future research could also consider transportation emissions as an extension of our findings.”

The “local scenario” could have been made more realistic by incorporating more economic factors in the simulations, and it could have benefited from further discussions of policies that would make locations of production and consumption of agricultural products closer to each other. There are some trade-offs between reducing GHG emissions and losses in economic efficiency. These losses could stem from economic situations that are not consistent with the prevalence of comparative advantage, where comparative advantage implies that one country produces a good or service for a lower opportunity cost than other countries.

We agree with the reviewer’s comment. We figured that there were too many assumptions involved in our “local production” scenario that distracted the main story of the paper, i.e., to estimate trade-adjusted agriculture emissions within the farm gate. Therefore, we removed this section on “local production” scenario. Instead, we included an additional section that focuses on the sensitivity analysis of our study.

The actual trends in international trade are largely determined by economic, geo-economic and political forces, and altering these patterns should be discussed in the context of reducing GHG emissions. Also, this study would benefit from a brief discussion of “optimal” outcomes that jointly encompass the economic factors (or economic welfare) and the objective of reducing GHG emissions.

We understand this comment to be linked to the first comment (see above), which is why we have decided to give an answer to this comment already above in the answer to that first comment. We entirely agree that current trade is determined by the forces and factors you mention. In the context of reducing GHG emissions, it is certainly the intention to do this in an optimal way, however this is defined. If we define it as cost-efficient, we would like to give incentives to produce agricultural and food products in regions with the lowest emissions intensity, or at least make emissions intensity one of various key factors in comparative advantage. Our trade-adjusted accounting approach would be suitable for reduction targets based on consumption-based emissions. As mentioned above, we have added the following paragraph as a conclusion.

“Considering TAEs in national emission inventories for the agricultural sector acknowledges demand as a driver for emissions. Under current climate change mitigation frameworks, national or regional commitments focus on production-based emissions, with demand playing only a minor role. As a result, this could lead to a displacement of emissions overseas. For example, stricter emission reduction targets for the European Union have been found to lead to a considerable displacement of saved emissions abroad, where agricultural emission intensities are higher ⁵⁵. Our analysis in calculating trade-adjusted emissions in the agricultural sector could serve as an important element towards constructing consumption-based national or regional emission reduction targets. In combination with trade policy regulations, such as border adjustment measures allowed by World Trade Organization regulations, such consumption-based targets would provide incentives to consider emissions intensities in agricultural production and trade, consequently leading to more cost-effective emission reductions overall.”

Some details about the local scenario are required. For instance, it is indicated that, through this scenario, there are reductions in imports and increases in local production to meet consumption needs. It is not clear how the latter is determined through the analysis. Also, it is noted that a country increases local production and decreases imports if it was already producing this commodity. Is there a certain production threshold for “already producing” or any minimal production amount is considered?

The second assumption is not clear, and its association with the note - that many countries still face several environmental and socio-economic barriers towards increasing productivity and achieving potential crop yields - needs further clarification. Also, I wonder if it is possible to somehow proxy such barriers in the simulation model.

We agree with the reviewer that our “local scenario” consists of a lot of assumptions. These assumptions distracted the main story of the paper, i.e., to estimate trade-adjusted agriculture emissions. Therefore, we remove this section on “local production” scenario. Instead, we included an additional section that focuses on the sensitivity analyses of our study.

Kander et al. (2015) propose an improvement to the consumption-based carbon accounting that takes technology differences in exporting sectors into consideration. Do you expect that this inclusion would impact your results and conclusion?

Many thanks for suggesting this remarkably interesting paper by Kander et al. (2015). Following the comment, we tested the method proposed by Kander et al. (2015) and re-estimated our emissions using this 'technology-adjustment' accordingly. We considered this test as a sensitivity analysis, and included the results in our revised manuscript as follows:

"In the second analysis, we adopted a 'technology-adjusted' approach in calculating TAE to account for technological differences in carbon efficiencies in the exports of different countries¹⁹. This approach has been proposed as an improvement to the conventional consumption-based accounting by more accurately reflecting how changes to national climate policies could affect overall global emissions¹⁹. Similar to the first analysis, the difference in switching to this approach was minor, with a mean decrease in trade-adjusted emissions of -2.4% from the original approach (Supplementary Fig. 8). However, the technology-adjusted approach did produce some interesting patterns at the regional and food group-specific levels (Supplementary Figs. 8, 9). For example, Europe and North America had lower TAEs for ruminant meat under this approach, reflecting the higher efficiency of ruminant meat production in these regions²⁰. We thus conclude that while our results were relatively stable even when incorporating a technology-adjusted approach, using such an approach also revealed several other noteworthy results, while at the same time reconfirming some of our earlier findings."

Please also see Supplementary Figures 7-8, which illustrate the differences between our original estimations and the 'technology-adjusted' approach. As quoted above, the overall difference is very small, but this sensitivity analysis has revealed some interesting patterns at the regional and food group-specific levels, which have also confirmed our earlier results.

Finally, I wonder about the role of (horizontal) Foreign Direct Investment (FDI) in bringing production and consumption locations closer to each other in some cases, as analyzed through the proximity-concentration trade-off framework (Brainard, 1997; Ghazalian and Furtan, 2009). International trade can be substituted in this case by local production of foreign affiliates of multinational enterprises. The author(s) could add a corresponding note in this context.

Many thanks for this interesting idea. We have now included a short discussion on reconnecting producers and consumers (i.e., bringing them closer) in terms of emissions associated with food transport as follows:

"Although food transport reduction through the regionalisation of food systems is an important climate change mitigation option⁴³, transport constitutes a very small share of overall food system emissions^{5,6,39,44}. Therefore, excluding its emissions would not make much difference in our results. Nevertheless, bringing producers and consumers closer through regionalised food systems would lead to more responsible food production and consumption⁴⁵, thus future research could also consider transportation emissions as an extension of our findings."

However, since we do not investigate the inputs and investment for food production in our manuscript, we did not discuss about FDI in the revised version. We are afraid that mentioning FDI would open a new dimension of discussion, which is beyond the scope of our analysis.

References

- Brainard, S.L., 1997. An Empirical Assessment of the Proximity-Concentration Trade-off between Multinational Sales and Trade. *American Economic Review*, 87(4), pp.520-544.
- Cristea, A., Hummels, D., Puzello, L. and Avetisyan, M., 2013. Trade and the Greenhouse Gas Emissions from International Freight Transport. *Journal of Environmental Economics and Management*, 65(1), pp.153-173.
- Ghazalian, P.L., 2017. The Effects of NAFTA/CUSFTA on Agricultural Trade Flows: An Empirical Investigation. *Canadian Journal of Agricultural Economics*, 65(2), pp.219-248.

Ghazalian, P.L. and Furtan, W.H., 2009. CUSFTA Effects: A Joint Consideration of Trade and Multinational Activities. *The Journal of International Trade & Economic Development*, 18(4), pp.487-504.

Kander, A., Jiborn, M., Moran, D.D. and Wiedmann, T.O., 2015. National Greenhouse-Gas Accounting for Effective Climate Policy on International Trade. *Nature Climate Change*, 5(5), pp.431-435.

Olper, A. and Raimondi, V., 2008. Agricultural Market Integration in the OECD: A Gravity-Border Effect Approach. *Food Policy*, 33(2), pp.165-175.

Sun, L. and Reed, M.R., 2010. Impacts of Free Trade Agreements on Agricultural Trade Creation and Trade Diversion. *American Journal of Agricultural Economics*, 92(5), pp.1351-1363.

Reference

Boehm et al. (2018). Comprehensive Life Cycle Assessment of Greenhouse Gas Emissions from U.S. Household Food Choices. *Food Policy*, 79, 67-76.

Cristea et al. (2011). Trade and the greenhouse gas emissions from international freight transport. NBER Working Paper Series.

Fellmann et al. (2018). Major challenges of integrating agriculture into climate change mitigation policy frameworks. *Mitigation and Adaptation Strategies for Global Change*, 23, 451-468.

Kander et al. (2015). National greenhouse-gas accounting for effective climate policy on international trade. *Nature Climate Change*, 5, 431-435.

Niles et al. (2018). Climate change mitigation beyond agriculture: a review of food system opportunities and implications. *Renewable Agriculture and Food Systems*, 33, 297-308.

Poore and Nemecek (2018). Reducing food's environmental impacts through producers and consumers. *Science*, 360, 987-992.

Sandström et al. (2018). The role of trade in the greenhouse gas footprints of EU diets. *Global Food Security*, 19, 48-55.

Vermeulen et al. (2012). Climate Change and Food Systems. *Annual Review of Environment and Resources*, 37, 195-222.

Reviewer #3

- What are the noteworthy results?

This manuscript provides consumption-based, national-level estimates of greenhouse gas emissions from the farming stage of human-food items. Consumption is evaluated as production plus imports minus exports. These three food quantities, as well as the emission intensities of the food products, are all taken from the public FAOSTAT website, a global, national-level database.

The authors provide the results of this calculation for the years 1987 to 2015.

The authors also compare these 'observed' consumption-based emissions with those in a hypothetical 'local production' scenario.

We thank the reviewer for the constructive comments, and for clearly and concisely pointing out issues that warrant our immediate attention for the revision, all of which have now been addressed.

- Does the work support the conclusions and claims, or is additional evidence needed?

- Are there any flaws in the data analysis, interpretation and conclusions? - Do these prohibit publication or require revision?

There is no obvious technical flaw I can find.

The 'local production' scenario is not meant to be realistic as the only constraint applied is that the country produces the crop already - for example, land availability, costs and other resources availability are not considered. So this is the same approach taken when calculated 'land savings' or 'water savings'. This is useful but has recognised and important limitations, such as its lack of practical meaning.

We agree with the reviewer that the 'local production' scenario does not provide much practical significance due to the very hypothetical and simplified nature of our assumption, in which we omitted considering resource availability, costs and economic benefits into this scenario. After careful consideration, and in light of other changes that have been made in the manuscript, we have decided to remove the 'local production' scenario from the revised version. Instead, we included an additional section that focuses on the sensitivity analyses of our study.

- Is the methodology sound? Does the work meet the expected standards in your field?

The methods overall are sound but quite simplistic relative to other studies in this field. Indeed, previous studies have implemented more sophisticated methods, which e.g. correct for most of re-exports, use product-specific emission intensities (rather than categories), design realistic counterfactual scenarios, among other advantages.

Our methodology is admittedly simple in terms of the assumptions and mathematics used, but we believe that this is an advantage as it is easily replicable by others to estimate, consistently and soundly, trade-adjusted agricultural emissions at the range of countries and food items that we strived for.

We do acknowledge that there are other studies that have used more "sophisticated methods", but these have not, to the best of our knowledge, been applied to the same scale of food items and countries that our study achieved. In addition, these other methods have certain limitations. Here we would like to highlight some examples:

- **The issue of re-exporters in trade flow analyses has been dealt with using a matrix algebra approach by Kastner et al. (2011). This approach has been applied in other studies analysing greenhouse gas emissions of food systems, although these were limited to specific countries (e.g., de Ruiter et al. 2016 for the United Kingdom), regions (e.g., Sandstrom et al. 2018 for the European Union; Schierhorn et al. 2019 for the former Soviet Union), and commodities (e.g., Henders et al. 2015 for 4 items, namely beef, soybean, palm oil, and wood products). We are not aware of this approach being applied to the range of items, years, and geographical scales that our study aimed for (i.e., 153 primary agricultural items, 30 years, and for all countries). Although we did not apply this matrix algebra approach, our study nonetheless has provided an overview of trade-adjusted agricultural emissions at the above-mentioned scale, which is currently lacking in the literature. However, considering the potential for this approach to be**

used for our study's context, we think that future research could apply this approach as a next step to address and investigate the issue of re-exporters.

- MRIOs are also another widely adopted method used to account for re-exporters. In preparation for the revision of this manuscript, we had a brief exchange with an expert on MRIO analyses, Dr. Martin Bruckner from the Vienna University of Economics and Business, on this issue. According to him, re-exporters are dealt with differently by the various MRIO databases, and a consistent approach in doing so is currently lacking. However, he did point out that a new version of an input-output model is currently being developed to tackle the issue of re-exporters in a systematic way that allows trade flows to be traced correctly.
- The use of product-specific emission intensities is indeed an important factor in input-output analyses. According to Steen-Olsen et al. (2014), aggregate CO₂ product multipliers (which is synonymous to our study's "emission intensity" for the "others" food group) can vary considerably depending on the level of sector (or food group) detail used to estimate the multiplier. However, this variability is less important for national-level total carbon footprint analyses such as ours, as the results of such analyses are "determined not from multipliers alone, but as the product of multipliers and consumption volumes." Steen-Olsen et al. (2014) therefore conclude that, for such analyses, "a higher level of detail will have some, but probably more limited, influence." In light of this, we conclude that our method of aggregating some of the food items in our dataset, i.e., "others", using the same coefficient is sufficiently sound, and that going into further detail may not yield much added value.

Furthermore, we conducted two separate sensitivity analyses which consider different forms of emission intensities. Besides highlighting the robustness of our results, our sensitivity analyses have also revealed several interesting results from the same dataset that also reconfirm some of our earlier findings.

- Is there enough detail provided in the methods for the work to be reproduced?

Yes.

Thank you very much for highlighting that our method is reproducible.

- Will the work be of significance to the field and related fields? How does it compare to the established literature? If the work is not original, please provide relevant references.

In my view, this manuscript is not innovative or significant enough to warrant a publication in Nature Communications.

We disagree with the reviewer comments that our manuscript is not innovative or significant enough. To address this comment, we highlighted key novelties of our study in the discussion section, with comparison to past studies that also investigated (agricultural) GHG emissions using a similar 'trade-adjustment' approach. In our revised manuscript, we further clarified these novelties, examples of which are as follows:

"To do this, we used a novel and consistent BTIO approach in which we explored all countries and all food items available from FAOSTAT. This approach extends previous studies that have also used a similar 'trade-adjustment' approach to estimate emissions, but these studies have so far limited their analyses to specific food items or geographical areas ¹⁶⁻¹⁸."

"We also show that both the production-based and trade-adjusted approaches towards emissions accounting produce widely differing results, particularly for major agricultural importers and exporters. While this difference has already been discussed in other studies for all sectors collectively ^{7,9,11-14}, our findings go one step deeper by exploring these differences specifically within the agricultural sector."

The main innovation of our study is to provide the trade-adjusted emissions for the agriculture and food sector at country scale. The IPCC Special Report on Climate Change and Land states that "Knowledge gaps include food consumption-based emissions at national scales, embedded emissions (overseas footprints) of food systems, comparison of GHG emissions per type of food systems (e.g., smallholder

and large-scale commercial food systems), and GHG emissions from land-based aquaculture.” (Please see Chapter 5, Section 5.7.5.2). Our study is a step forward to close this knowledge gap.

The concepts, approaches and assumptions are not new. Indeed many studies that are cited here in the references refer to similar work: Davis and Caldeira (ref.3) and Caro et al. (ref.4) on the importance of consumption-based accounting of GHG emissions; Caro et al. (ref. 25) on emissions embodied in meat trade; Watts et al. 2019 (not cited; reference: The 2019 report of The Lancet Countdown on health and climate change, The Lancet, 394, 10211, 2019, [https://doi.org/10.1016/S0140-6736\(19\)32596-6](https://doi.org/10.1016/S0140-6736(19)32596-6).) also provides the production-based GHG emissions of food products over time by country, but using a process-based model instead of the FAOSTAT data for emission intensity estimates.

We agree that the concepts behind our study (for e.g., the BTIO approach towards ‘consumption-based’ emissions accounting) are not new, and have indeed been widely explored in past studies. However, the application of this approach specifically to the agricultural sector at our study’s level of food item-specificity and geographical coverage has, to the best of our knowledge, not been attempted. To clarify this, and to show the distinction of our study from past research, we would like to comment on each literature mentioned by the reviewer above in Table 1. We have also included other studies that have used a similar BTIO approach, i.e., adjusting national emissions with bilateral trade:

Table 1: List of papers and their respective comments.

Paper	Comment
Caro et al. (2017): Mapping the international flows of GHG emissions within a more feasible consumption-based framework.	Because the study relied on World Bank emissions data, it excluded non-CO2 GHG emissions (e.g., methane and nitrous oxides). This is in contrast to our study in which we included these other non-CO2 GHG emissions because of our use of FAOSTAT data. In addition, and most importantly, the agriculture sector is a major emitter of non-CO2 GHG emissions.
Caro et al. (2014): CH4 and N2O emissions embodied in international trade of meat.	As mentioned in the title, the study focused only on emissions embodied in meat (specifically beef, pork, and chicken), whereas our study aimed to estimate emissions for all food groups.
Watts et al. (2019): The 2019 report of The Lancet Countdown on health and climate change: ensuring that the health of a child born today is not defined by a changing climate.	The agricultural emissions estimated in this study only covered the production stage, and did not include bilateral trade adjustments or consumption-based estimations.
Schierhorn et al. (2019): Large greenhouse gas savings due to changes in the post-Soviet food systems.	The study only focused on emissions associated with the food system of the former Soviet Union.
Davis & Caldeira (2010): Consumption-based accounting of CO2 emissions.	The consumption-based emissions estimated in these studies covered all sectors, and were not specific to agricultural commodity or food group.
Peters and Hertwich (2008): CO2 Embodied in International Trade with Implications for Global Climate Policy.	

Peters et al. (2011): Growth in emission transfers via international trade from 1990 to 2008.	
---	--

Besides, the approach in terms of data collection/combination is also simple because all the data used in this manuscript comes from the same source; this has the advantage of (hopefully) inherent consistency but means that besides the calculation, there is no value added in terms of combining different datasets.

We see our approach of using data from the same source (i.e., FAOSTAT) as an advantage rather than a disadvantage. As the reviewer already pointed out, this 'single data source' approach should ensure inherent consistency. We did not incorporate data from other sources that could have increased the coverage of our study to include more items (e.g., fish and aquaculture products) or other stages of the food system to avoid (potentially) introducing more uncertainties into our analysis from the use of different datasets.

Overall, although it is interesting and an important issue, I believe the presented work is too incremental.

I suggest the authors significantly change the design of the study to make a more substantial addition to the literature.

We share the reviewer's comment about the interest and importance of the issues covered by our study. However, we do not agree that our work is "too incremental". On the contrary, as mentioned previously, our study is a major step forward in closing the knowledge gap on food consumption-based emissions at national levels, a gap that was highlighted in the IPCC Special Report on Climate Change (see Chapter 5, Section 5.7.5.2). Furthermore, as highlighted in Table 1 above, no other study to our knowledge has applied this 'trade-adjustment' approach to national agricultural emissions, at the range of countries, years, and food items simultaneously as we did in our study.

Nonetheless, we welcome any suggestions on how we could improve our study's concepts and approach. Indeed, other reviewers have highlighted the importance in differentiating the different methods of conducting a consumption-based analyses (i.e., a Bilateral Trade Input-Output (BTIO) approach vs. a Multi-Regional Input-Output (MRIO) approach), or the potential implications of considering technology differences in the export sectors of different countries (see Kander et al., 2015).

These considerations have been taken into account in the revision of our manuscript, and we have even included two sensitivity analyses in that regard. Additionally, we conducted a simple test on how consideration of agricultural land use emissions would change trade-adjusted agricultural emissions in this revised version. However, it was not clear to us what changes were expected specifically by the reviewer, nor how our methodology could have been improved from the reviewer's perspective. Nonetheless, we are confident that the current changes have significantly improved our methodology and clarified our study's objectives, and we hope that these changes have addressed the reviewer's concerns.

References

Caro et al. (2014). CH₄ and N₂O emissions embodied in international trade of meat. *Environmental Research Letters*, 9.

Caro et al. (2017). Mapping the international flows of GHG emissions within a more feasible consumption-based framework. *Journal of Cleaner Production*, 147, 142-151.

Davis and Caldeira (2010). Consumption-based accounting of CO₂ emissions. *Proceedings of the National Academy of Sciences of the United States of America*, 107, 5687-5692.

Kander et al. (2015). National greenhouse-gas accounting for effective climate policy on international trade. *Nature Climate Change*, 5, 431-435.

Mbow et al. (2019). Climate Change and Land: an IPCC special report on climate change, desertification, land degradation, sustainable land management, food security, and greenhouse gas fluxes in terrestrial ecosystems (eds. Shukla et al.) 437–550.

Peters and Hertwich (2008). CO₂ embodied in international trade with implications for global climate policy. *Environmental Science and Technology*, 42, 1401-1407.

Peters et al. (2011). Growth in emission transfers via international trade from 1990 to 2008. *Proceedings of the National Academy of Sciences of the United States of America*, 108, 8903-8908.

Schierhorn et al. (2019). Large greenhouse gas savings due to changes in the post-Soviet food systems. *Environmental Research Letters*, 14, 65009.

Watts et al. (2019). The 2019 report of The Lancet Countdown on health and climate change: ensuring that the health of a child born today is not defined by a changing climate. *The Lancet*, 394:10211, 1836-1878.

Peer Review File

Reviewer comments, second round review –

Reviewer #1 (Remarks to the Author):

Dear authors,

The MS "Adjusting agricultural emissions for trade matters for climate change mitigation" by Foong, Pradhan and co-authors that I previously revised for potential publication in Nature Communications has been reviewed again taking into consideration the responses to my previous comments made by authors.

I would like to highlight that the work has been substantially improved with a huge effort you made. I recognize that my comments and suggested revision were substantial and implied a significant intellectual effort. I do appreciate the dedication of authors to address all comments. Mainly the issues related to the role of LUC in your analysis and further assessment have been properly revised and the text was refined accordingly. Based on all this, I do consider that the MS is suitable for publication in Nature Communications in its current form.

Yours sincerely,

Alberto Sanz-Cobena

Reviewer #2 (Remarks to the Author):

Adjusting Agricultural Emissions for Trade Matters for Climate Change Mitigation

(Manuscript ID: NCOMMS-20-24771A)

The authors have successfully addressed my previous comments. I have no further remarks.

Reviewer #3 (Remarks to the Author):

The authors have carefully responded to the reviewers' comments. However I still have major concerns described below, which mainly pertain to the significance of the contribution to the literature. The content is otherwise fairly presented and the methodology is sound. My concerns relate to the scope and approach chosen. I am adding here more specific points to my last review.

Comments:

- The methods section reads "we assumed exports to come entirely from the exporter's production, or, if the exporter does not produce the item, entirely from the exporter's immediate region."

As the authors recognised, this is an "important assumption". More importantly, this assumption brings strong limitations to the methodology used, because the set goal of this work - as far as I understand- is to show how accounting of GHG emissions from farming differs between common

production-based approaches and a consumption-based (or trade-adjusted) approach. Two variables are playing a role in this difference: (1) the trade pattern, i.e. how much of the food consumed nationally is produced abroad (what the author refer to as 'volumes') and (2) the different technologies, or emissions intensities (EI), of food products between countries. Because the authors are interested in GHG emissions from the production stage / farm stage, it is crucial to know the origin of the product in order to use the appropriate emission intensity associated with food production.

If not (which is the authors' current approach and was kept in the revised version), the role played by the EI in the difference between production and consumption-based approaches is not properly accounted for, due to re-exports.

If the authors wish to maintain this approach, it would be highly valuable to (i) further clarify the limitation of this assumption, including by using results from previous literature about the importance of re-exports to estimate the uncertainty brought by this assumption (e.g. Kastner et al. 2014) and, (ii) importantly, to add insights by extensively discussing the comparison of emission accounting results, by country, between the main approach (TAE), a production-based approach (as briefly mentioned I.288); and the role played by simply adding trade pattern to the production-based approach (e.g. allocating a global mean EI by product to imports - that is similar to the first 'sensitivity analysis' but with an average EI for all trade partner countries) and then by also considering a spatially-explicit EI.

- As mentioned in the previous round of reviews, similar studies have been published, with two additional advantages: (i) considering re-exports in order to use the appropriate EI in the emissions embedded in trade and (ii) using product-specific EI which avoid commodities aggregation in large groups. Notably, Watts et al. (2020) estimated, with a global coverage of crop and livestock commodities over a long time period (20 years) and at the global scale, both production-based GHG emissions and those with adjusted trade and corrected re-exports. Watts et al.: The 2020 report of The Lancet Countdown on health and climate change: responding to converging crises; [https://www.thelancet.com/journals/lancet/article/PIIS0140-6736\(20\)32290-X/fulltext](https://www.thelancet.com/journals/lancet/article/PIIS0140-6736(20)32290-X/fulltext)

- In the response to reviewers, I found this statement confusing: "Regarding emissions due to livestock feed, we have implicitly considered the emissions associated with feed in our estimations, as our trade-matrix includes crops whose by-products are used as livestock feed (e.g., cake of groundnuts, sesame, and soybeans; see also Supplementary Table 2). Furthermore, the production of feed (and emissions associated with it) is considered a pre-production process (Vermeulen et al. 2012). As our study primarily focuses on the agricultural production stage (i.e., farm-gate stage), we have decided to exclude feed as a separate, unique process in estimating emissions related to livestock, similar to our exclusion of LUC-related emissions from our study's main focus."

Did the authors include emissions from feed ("we have implicitly considered") or not ("we have decided to exclude feed as a separate, unique process in estimating emissions related to livestock") ? Please clarify in the text.

- "Nevertheless, bringing producers and consumers closer through regionalised food systems would lead to more responsible food production and consumption 45,"

I would say this is not an established truth. I would change the end of the sentence "would lead" to "could lead". Also because "responsible" can be interpreted in many ways.

REVIEWER COMMENTS

Reviewer #1 (Remarks to the Author):

Dear authors,

The MS "Adjusting agricultural emissions for trade matters for climate change mitigation" by Foong, Pradhan and co-authors that I previously revised for potential publication in Nature Communications has been reviewed again taking into consideration the responses to my previous comments made by authors.

I would like to highlight that the work has been substantially improved with a huge effort you made. I recognize that my comments and suggested revision were substantial and implied a significant intellectual effort. I do appreciate the dedication of authors to address all comments. Mainly the issues related to the role of LUC in your analysis and further assessment have been properly revised and the text was refined accordingly. Based on all this, I do consider that the MS is suitable for publication in Nature Communications in its current form.

Yours sincerely,

Response: We thank the reviewer for dedicating valuable time to review our manuscript. We are pleased to know that the reviewer is satisfied with our revised manuscript.

Reviewer #2 (Remarks to the Author):

Adjusting Agricultural Emissions for Trade Matters for Climate Change Mitigation

Nature Communications (Manuscript ID: NCOMMS-20-24771A)

The authors have successfully addressed my previous comments. I have no further remarks.

Response: We thank the reviewer for dedicating valuable time to review our manuscript. We are pleased to know that the reviewer is satisfied with our revised manuscript.

Reviewer #3 (Remarks to the Author):

The authors have carefully responded to the reviewers' comments. However, I still have major concerns described below, which mainly pertain to the significance of the contribution to the literature. The content is otherwise fairly presented and the methodology is sound. My concerns relate to the scope and approach chosen. I am adding here more specific points to my last review.

Response: We are pleased that the reviewer found our study's content, methodology, and responses to be fairly presented and sound. We also highly appreciate the reviewer for raising her/his concerns regarding the novelty and approach of our study. These concerns have now been taken into full consideration in the latest round of changes.

Comments:

- The methods section reads "we assumed exports to come entirely from the exporter's production, or, if the exporter does not produce the item, entirely from the exporter's immediate region."

As the authors recognised, this is an "important assumption". More importantly, this assumption brings strong limitations to the methodology used, because the set goal of this work - as far as I understand - is to show how accounting of GHG emissions from farming differs between common production-based approaches and a consumption-based (or trade-adjusted) approach.

Two variables are playing a role in this difference: (1) the trade pattern, i.e. how much of the food consumed nationally is produced abroad (what the author refer to as 'volumes') and (2) the different technologies, or emissions intensities (EI), of food products between countries.

Because the authors are interested in GHG emissions from the production stage / farm stage, it is crucial to know the origin of the product in order to use the appropriate emission intensity associated with food production.

If not (which is the authors' current approach and was kept in the revised version), the role played by the EI in the difference between production and consumption-based approaches is not properly accounted for, due to re-exports.

If the authors wish to maintain this approach, it would be highly valuable to (i) further clarify the limitation of this assumption, including by using results from previous literature about the importance of re-exports to estimate the uncertainty brought by this assumption (e.g. Kastner et al. 2014) and, (ii) importantly, to add insights by extensively discussing the comparison of emission accounting results, by country, between the main approach (TAE), a production-based approach (as briefly mentioned I.288); and the role played by simply adding trade pattern to the production-based approach (e.g. allocating a global mean EI by product to imports - that is similar to the first 'sensitivity analysis' but with an average EI for all trade partner countries) and then by also considering a spatially-explicit EI.

Response: We would like to thank the reviewer very much for these constructive and insightful comments. Based on the reviewer's suggestions, we have now included an additional third sensitivity analysis to account for re-exports (see revised text and Supplementary Figs. 9 and 10). In short, our third sensitivity analysis involves recalculating emission intensities based on emissions from both production and imports. The precise method is explained in our methods section as follows:

"In the third sensitivity analysis, we estimated TAEs using emission intensities that account for both production and imports. To do this, we calculated, for each country, the sum of emissions from domestic production and imports, using the relevant emission intensities as provided by FAOSTAT. We then divided emissions with the sum of production and import volumes to obtain a new set of emission intensities. Using this approach, we were able to link country-specific production emissions with imports in each trade flow. This essentially differs from the assumption of the original analysis, as we now consider that countries export both domestically produced and imported items. One important assumption is that we assume exports to come in proportional shares from import and domestic production, due to the lack of detailed information on the shares that go into exports⁴⁷. Furthermore, the new set of emission intensities considers first-order trade (i.e., exports of partner countries' production), rather than second-order trade and beyond (i.e., exports of partner countries' imports)."

We found that our original results are robust to re-export considerations, as the differences between the 're-exporter' approach and our original approach were small on average. This finding is further elaborated in the results section as follows:

"As was the case with the previous sensitivity analyses, using this approach also produced very small differences. During the period of 1986-2017, we found that, for countries that saw an increase in TAE from the original approach, the mean change was +1.3%; for countries that saw a decrease in TAE, the mean change was -5.4% (Supplementary Fig. 9)."

Additionally, we have also acknowledged the limitation of our main assumption (i.e., assuming exports to come entirely from the exporter's production), by including the following statement in the discussion:

"Another point for consideration, which is particularly relevant for trade analyses, is our assumption that exports come entirely from each exporter's domestic production, which may not sufficiently account for re-exports. Including re-exports into emission calculations could alter results to varying degrees as compared to relying solely on reported trade links, which was our main approach ⁴⁷. To address this, we therefore included a sensitivity analysis which used emission intensities that account for both production and imports (see Supplementary Figure 9)."

- As mentioned in the previous round of reviews, similar studies have been published, with two additional advantages: (i) considering re-exports in order to use the appropriate EI in the emissions embedded in trade and (ii) using product-specific EI which avoid commodities aggregation in large groups. Notably, Watts et al. (2020) estimated, with a global coverage of crop and livestock commodities over a long time period (20 years) and at the global scale, both production-based GHG emissions and those with adjusted trade and corrected re-exports.

Watts et al.: The 2020 report of The Lancet Countdown on health and climate change: responding to converging crises; [https://www.thelancet.com/journals/lancet/article/PIIS0140-6736\(20\)32290-X/fulltext](https://www.thelancet.com/journals/lancet/article/PIIS0140-6736(20)32290-X/fulltext)

Response: We thank the reviewer for highlighting this paper by Watts et al. (2020), and we acknowledge the similarities of this paper with ours in terms of geographical, temporal, and commodity scope. We have now cited this paper in our revised manuscript accordingly.

However, we would like to emphasise one additional - and important - nuance that our paper offers. Our approach compares emissions at multiple geographical levels, including at - and with specific focus on - the country level. This, we believe, is highly relevant for trade policy negotiations and implementation, which are normally conducted at the national and bilateral levels. In our manuscript, we have highlighted our paper's important contribution in several parts of the discussion section, for example:

"Specifically, we calculated emissions within the farm-gate that are embodied in agricultural trade flows, and used these to estimate TAE at the country level, which we believe would have significant trade policy implications."

"... the BTIO approach is more transparent and suited for analysing bilateral political agreements as well as trade and climate policies ¹²."

"Our analysis in calculating trade-adjusted emissions in the agricultural sector could serve as an important element towards constructing consumption-based national or regional emission reduction targets."

Relatedly, and to bring home the relevance of our paper to recent and ongoing policy discussions over trade and carbon emissions, we have included in the concluding paragraph of the discussion section a reference to the recent proposal by the European Union for border adjustments on carbon emission-intensive commodities like steel, cement, aluminium, and fertilisers. This is because a trade-adjustment approach for food items may be an important element to extending such an adjustment also to the food sector.

Furthermore, our study tested various interpretations of 'emission factors' in estimating trade-adjusted agricultural emissions. This can be seen from our multi-faceted sensitivity analyses, which tested

different assumptions of emission factors, including with regards to technology-adjustments and re-exports. While the findings from these sensitivity analyses were largely similar to our main approach at the global and regional levels, the finer scale differences at the country and food-specific levels provide further insights into the links between the various interpretations of emission factors and the resulting emissions accounting at the national level.

Fittingly, we believe that such a study that covers the extensive range of issues associated with trade-adjusted agricultural emissions warrants a more detailed, in-depth analysis that also identifies research gaps and ways forward to address the topic. This is precisely what our paper aims to achieve, thereby filling an important gap in the literature with regards to ongoing discussions over agricultural emissions accounting and global climate action.

- In the response to reviewers, I found this statement confusing: “Regarding emissions due to livestock feed, we have implicitly considered the emissions associated with feed in our estimations, as our trade-matrix includes crops whose by-products are used as livestock feed (e.g., cake of groundnuts, sesame, and soybeans; see also Supplementary Table 2).

Furthermore, the production of feed (and emissions associated with it) is considered a pre-production process (Vermeulen et al. 2012). As our study primarily focuses on the agricultural production stage (i.e., farm-gate stage), we have decided to exclude feed as a separate, unique process in estimating emissions related to livestock, similar to our exclusion of LUC-related emissions from our study’s main focus.”

Did the authors include emissions from feed (“we have implicitly considered”) or not (“we have decided to exclude feed as a separate, unique process in estimating emissions related to livestock”)? Please clarify in the text.

Response: Thank you very much for highlighting this slight inconsistency regarding feed items. We would now like to clarify this issue as follows:

- We did not include emissions from feed *production* explicitly, as our study’s focus is on the agricultural production stage only. This is mentioned in the following paragraph in the discussion section:

“Furthermore, our study is mostly focused on emissions from the agricultural production stage, and does not consider emissions from other pre- and post-production processes...”

- However, our analysis does implicitly include emissions due to the production of crops that are used as feed, such as soybeans and cake of various crops (see Supplementary Table 2). This is because several items, such as soybeans, could be used either directly for human food consumption or as livestock feed.

Hence, we have now included an additional statement in the discussion section to further clarify the matter:

“Moreover, our analysis implicitly accounted for emissions due to the production of crops that are used as feed items, as several crops such as soybeans can be used both directly for human food consumption and as livestock feed (see Supplementary Table 2).”

We hope that the above explanations sufficiently address the reviewer’s questions.

- “Nevertheless, bringing producers and consumers closer through regionalised food systems would lead to more responsible food production and consumption 45,”

I would say this is not an established truth. I would change the end of the sentence “would lead” to “could lead”. Also because “responsible” can be interpreted in many ways.

Response: Thank you very much for your suggestion. We changed the end of the sentence accordingly.

Reviewer #3 (Remarks to the Author):

The authors have carefully responded to the reviewers' comments. However, I still have major concerns described below, which mainly pertain to the significance of the contribution to the literature. The content is otherwise fairly presented and the methodology is sound. My concerns relate to the scope and approach chosen. I am adding here more specific points to my last review.

Response: We are pleased that the reviewer found our study's content, methodology, and responses to be fairly presented and sound. We also highly appreciate the reviewer for raising her/his concerns regarding the novelty and approach of our study. These concerns have now been taken into full consideration in the latest round of changes.

Reviewer response: Thank you for responding to my comments. Very important corrections still need to be made in order to accurately present this work and to accurately place it within the published literature. Some of the corrections needed concern newly added text/references made in the revision. Please find more details below.

Comments:

- The methods section reads "we assumed exports to come entirely from the exporter's production, or, if the exporter does not produce the item, entirely from the exporter's immediate region."

As the authors recognised, this is an "important assumption". More importantly, this assumption brings strong limitations to the methodology used, because the set goal of this work - as far as I understand - is to show how accounting of GHG emissions from farming differs between common production-based approaches and a consumption-based (or trade-adjusted) approach.

Two variables are playing a role in this difference: (1) the trade pattern, i.e. how much of the food consumed nationally is produced abroad (what the author refer to as 'volumes') and (2) the different technologies, or emissions intensities (EI), of food products between countries.

Because the authors are interested in GHG emissions from the production stage / farm stage, it is crucial to know the origin of the product in order to use the appropriate emission intensity associated with food production.

If not (which is the authors' current approach and was kept in the revised version), the role played by the EI in the difference between production and consumption-based approaches is not properly accounted for, due to re-exports.

If the authors wish to maintain this approach, it would be highly valuable to (i) further clarify the limitation of this assumption, including by using results from previous literature about the importance of re-exports to estimate the uncertainty brought by this assumption (e.g. Kastner et al. 2014) and, (ii) importantly, to add insights by extensively discussing the comparison of emission accounting results, by country, between the main approach (TAE), a production-based approach (as briefly mentioned I.288); and the role played by simply adding trade pattern to the production-based approach (e.g. allocating a global mean EI by product to imports - that is similar to the first 'sensitivity analysis' but with an average EI for all trade partner countries) and then by also considering a spatially-explicit EI.

Response: We would like to thank the reviewer very much for these constructive and insightful comments. Based on the reviewer's suggestions, we have now included an additional third sensitivity analysis to account for re-exports (see revised text and Supplementary Figs. 9 and 10). In short, our third sensitivity analysis involves recalculating emission intensities based on emissions from both production and imports. The precise method is explained in our methods section as follows:

"In the third sensitivity analysis, we estimated TAEs using emission intensities that account for both production and imports. To do this, we calculated, for each country, the sum of emissions from domestic production and imports, using the relevant emission intensities as provided by FAOSTAT. We then divided emissions with the sum of production and import volumes to obtain a new set of emission intensities. Using this approach, we were able to link country-specific production emissions with imports in each trade flow. This essentially differs from the assumption of the original analysis, as we now consider that countries export both domestically produced and imported items. One important assumption is that we assume exports to come in proportional shares from import and domestic production, due to the lack of detailed information on the shares that go into exports⁴⁷. Furthermore, the new set of emission intensities considers first-order trade (i.e., exports of partner countries' production), rather than second-order trade and beyond (i.e., exports of partner countries' imports)."

We found that our original results are robust to re-export considerations, as the differences between the 're-exporter' approach and our original approach were small on average. This finding is further elaborated in the results section as follows:

"As was the case with the previous sensitivity analyses, using this approach also produced very small differences. During the period of 1986-2017, we found that, for countries that saw an increase in TAE from the original approach, the mean change was +1.3%; for countries that saw a decrease in TAE, the mean change was -5.4% (Supplementary Fig. 9)."

Reviewer response: The authors have here mainly reported mean differences. However, what is most relevant for the study's stated goals are country-level differences. Please provide range, median, etc., and extensively develop the comparisons between estimates at the country-level.

Additionally, we have also acknowledged the limitation of our main assumption (i.e., assuming exports to come entirely from the exporter's production), by including the following statement in the discussion:

"Another point for consideration, which is particularly relevant for trade analyses, is our assumption that exports come entirely from each exporter's domestic production, which may not sufficiently account for re-exports. Including re-exports into emission calculations could alter results to varying degrees as compared to relying solely on reported trade links, which was our main approach⁴⁷. To address this, we therefore included a sensitivity analysis which used emission intensities that account for both production and imports (see Supplementary Figure 9)."

Reviewer response: While the sensitivity analysis means the issue of re-exports is acknowledged, it is still too peripheral, despite trade accounting being the main focus of this study (cf. title). This point should become an integral part of the manuscript, rather than a third sensitivity analysis.

Given that both the 2020 and 2021 Lancet Countdown reports (Watts et al. 2020 and Romanello et al. 2021) provided estimates of agricultural emissions with a large product coverage, at the national scale, and over time – and for the 2021 report, both consumption-based (with re-export correction) and production-based accounts – this study's contribution is currently too thin.

An option could be to 1) explicitly consider re-exports in the analysis and 2) compare the results of several approaches, including that of Romanello et al. 2021. Approaches could include consumption-based following the authors' approach / Romanello vs production-based following the authors' approach / Romanello.

References:

- Watts et al.: The 2020 report of The Lancet Countdown on health and climate change: responding to converging crises; [https://www.thelancet.com/journals/lancet/article/PIIS0140-6736\(20\)32290-X/fulltext](https://www.thelancet.com/journals/lancet/article/PIIS0140-6736(20)32290-X/fulltext)

- Romanello et al.: The 2021 report of the Lancet Countdown on health and climate change: code red for a healthy future; [https://doi.org/10.1016/S0140-6736\(21\)01787-6](https://doi.org/10.1016/S0140-6736(21)01787-6).

- Data from Romanello et al. 2021: <https://www.lancetcountdown.org/data-platform/mitigation-actions-and-health-co-benefits/3-5-food-agriculture-and-health/3-5-1-emissions-from-agricultural-production-and-consumption>

- As mentioned in the previous round of reviews, similar studies have been published, with two additional advantages: (i) considering re-exports in order to use the appropriate EI in the emissions embedded in trade and (ii) using product-specific EI which avoid commodities aggregation in large groups. Notably, Watts et al. (2020) estimated, with a global coverage of crop and livestock commodities over a long time period (20 years) and at the global scale, both production-based GHG emissions and those with adjusted trade and corrected re-exports.

Watts et al.: The 2020 report of The Lancet Countdown on health and climate change: responding to converging crises; [https://www.thelancet.com/journals/lancet/article/PIIS0140-6736\(20\)32290-X/fulltext](https://www.thelancet.com/journals/lancet/article/PIIS0140-6736(20)32290-X/fulltext)

Response: We thank the reviewer for highlighting this paper by Watts et al. (2020), and we acknowledge the similarities of this paper with ours in terms of geographical, temporal, and commodity scope. We have now cited this paper in our revised manuscript accordingly.

However, we would like to emphasise one additional - and important - nuance that our paper offers. Our approach compares emissions at multiple geographical levels, including at - and with specific focus on - the country level.

Reviewer response: actually, the country-level analysis is also available in this published literature (cf. references above). Please adjust all references in the text accordingly – as suggested above, this is an opportunity to adjust the study in order for it to add more value to the literature on this topic.

This, we believe, is highly relevant for trade policy negotiations and implementation, which are normally conducted at the national and bilateral levels. In our manuscript, we have highlighted our paper's important contribution in several parts of the discussion section, for example:

“Specifically, we calculated emissions within the farm-gate that are embodied in agricultural trade flows, and used these to estimate TAE at the country level, which we believe would have significant trade policy implications.”

“... the BTIO approach is more transparent and suited for analysing bilateral political agreements as well as trade and climate policies ¹².”

“Our analysis in calculating trade-adjusted emissions in the agricultural sector could serve as an important element towards constructing consumption-based national or regional emission reduction targets.”

Relatedly, and to bring home the relevance of our paper to recent and ongoing policy discussions over trade and carbon emissions, we have included in the concluding paragraph of the discussion section a reference to the recent proposal by the European Union for border adjustments on carbon emission-intensive commodities like steel, cement, aluminium, and fertilisers. This is because a trade-adjustment approach for food items may be an important element to extending such an adjustment also to the food sector.

Reviewer: great. That may help support more added value to this study if different approaches are more extensively compared.

Furthermore, our study tested various interpretations of 'emission factors' in estimating trade-adjusted agricultural emissions. This can be seen from our multi-faceted sensitivity analyses, which tested

different assumptions of emission factors, including with regards to technology-adjustments and re-exports. While the findings from these sensitivity analyses were largely similar to our main approach at the global and regional levels, the finer scale differences at the country and food-specific levels provide further insights into the links between the various interpretations of emission factors and the resulting emissions accounting at the national level.

Fittingly, we believe that such a study that covers the extensive range of issues associated with trade-adjusted agricultural emissions warrants a more detailed, in-depth analysis that also identifies research gaps and ways forward to address the topic. This is precisely what our paper aims to achieve, thereby filling an important gap in the literature with regards to ongoing discussions over agricultural emissions accounting and global climate action.

- In the response to reviewers, I found this statement confusing: “Regarding emissions due to livestock feed, we have implicitly considered the emissions associated with feed in our estimations, as our trade-matrix includes crops whose by-products are used as livestock feed (e.g., cake of groundnuts, sesame, and soybeans; see also Supplementary Table 2).

Furthermore, the production of feed (and emissions associated with it) is considered a pre-production process (Vermeulen et al. 2012). As our study primarily focuses on the agricultural production stage (i.e., farm-gate stage), we have decided to exclude feed as a separate, unique process in estimating emissions related to livestock, similar to our exclusion of LUC-related emissions from our study’s main focus.”

Did the authors include emissions from feed (“we have implicitly considered”) or not (“we have decided to exclude feed as a separate, unique process in estimating emissions related to livestock”)? Please clarify in the text.

Response: Thank you very much for highlighting this slight inconsistency regarding feed items. We would now like to clarify this issue as follows:

- We did not include emissions from feed *production* explicitly, as our study’s focus is on the agricultural production stage only. This is mentioned in the following paragraph in the discussion section:

“Furthermore, our study is mostly focused on emissions from the agricultural production stage, and does not consider emissions from other pre- and post-production processes...”

- However, our analysis does implicitly include emissions due to the production of crops that are used as feed, such as soybeans and cake of various crops (see Supplementary Table 2). This is because several items, such as soybeans, could be used either directly for human food consumption or as livestock feed.

Hence, we have now included an additional statement in the discussion section to further clarify the matter:

“Moreover, our analysis implicitly accounted for emissions due to the production of crops that are used as feed items, as several crops such as soybeans can be used both directly for human food consumption and as livestock feed (see Supplementary Table 2).”

We hope that the above explanations sufficiently address the reviewer’s questions.

Reviewer response: OK. You may simply say that feed crops are allocated to crop accounts rather than livestock accounts. This is another issue for trade that is worth mentioning (as feed crops can be traded before the livestock products may be trade themselves).

- “Nevertheless, bringing producers and consumers closer through regionalised food systems would lead to more responsible food production and consumption 45,”

I would say this is not an established truth. I would change the end of the sentence “would lead” to “could lead”. Also because “responsible” can be interpreted in many ways.

Response: Thank you very much for your suggestion. We changed the end of the sentence accordingly.

Rebuttal letter

REVIEWER COMMENTS

Reviewer #3 (Remarks to the Author):

Thank you for responding to my comments. Very important corrections still need to be made in order to accurately present this work and to accurately place it within the published literature. Some of the corrections needed concern newly added text/references made in the revision. Please find more details below.

Response: We would like to thank the reviewer again for another round of very constructive comments and suggestions for our manuscript. We hope that we now addressed all concerns in the revised manuscript, with our specific responses outlined below.

Reviewer response: The authors have here mainly reported mean differences. However, what is most relevant for the study's stated goals are country-level differences. Please provide range, median, etc., and extensively develop the comparisons between estimates at the country-level text.

Response: We would like to thank the reviewer for pointing this out. We have included several explanations in the revised version of the results to provide a better overview of the distribution of the estimates, as well as the most extreme differences between our main approach and the sensitivity analyses. For the sensitivity analysis, for example, we have included the following explanation:

“The largest differences were consistently found in several small-island and import-dependent countries where differences in emission intensities are more apparent in the resulting TAEs due to the larger share of traded products in their inventories. This was most notably the case for Kiribati, which saw the largest difference of almost 19 times in 2000. These extreme values, however, only accounted for a small share of differences during the period of 1986-2017, with over 98% of differences falling within +/-10% from the original approach.”

Additionally, we have also presented these differences graphically in the supplementary information as maps (see Supplementary Figures 7, 8, 9).

Reviewer response: While the sensitivity analysis means the issue of re-exports is acknowledged, it is still too peripheral, despite trade accounting being the main focus of this study (cf. title). This point should become an integral part of the manuscript, rather than a third sensitivity analysis.

Given that both the 2020 and 2021 Lancet Countdown reports (Watts et al. 2020 and Romanello et al. 2021) provided estimates of agricultural emissions with a large product coverage, at the national scale, and over time – and for the 2021 report, both consumption-based (with re-export correction) and production-based accounts – this study's contribution is currently too thin.

An option could be to 1) explicitly consider re-exports in the analysis and 2) compare the results of several approaches, including that of Romanello et al. 2021. Approaches could include consumption-based following the authors' approach / Romanello vs production-based following the authors' approach / Romanello.

References:

- Watts et al.: The 2020 report of The Lancet Countdown on health and climate change: responding to converging crises; [https://www.thelancet.com/journals/lancet/article/PIIS0140-6736\(20\)32290-X/fulltext](https://www.thelancet.com/journals/lancet/article/PIIS0140-6736(20)32290-X/fulltext)
- Romanello et al.: The 2021 report of the Lancet Countdown on health and climate change: code red for a healthy future; [https://doi.org/10.1016/S0140-6736\(21\)01787-6](https://doi.org/10.1016/S0140-6736(21)01787-6).

Data from Romanello et al. 2021: <https://www.lancetcountdown.org/data-platform/mitigationactions-and-health-co-benefits/3-5-food-agriculture-and-health/3-5-1-emissions-from-agriculturalproductionand-consumption>.

Response: We appreciate the reviewer's suggestions on how our manuscript could better incorporate the issue of re-exports into our findings and discussion. After deliberating on the various approaches to consider trade (and relatedly, re-exports) in emissions accounting, we have decided to stick to our original approach. Below are the reasons for our decision:

- We believe our original approach, which uses FAOSTAT data as they are reported, brings into the discussion the roles that intermediary trading countries play in overall emissions reductions. While the approach used by Romanello et al. (2021), which uses the methodology introduced by Kastner et al. (2021), does attempt to correct for re-exports, it is fundamentally to take a consumption-based perspective, which we believe does not sufficiently account for the role of intermediary trading countries. This consideration is important because trade also provides economic benefits to re-exporting countries. With re-export correction, as their roles in reducing emissions associated with trade is somewhat blurred or hidden. In our approach, a certain proportion of emissions due to trade is allocated to re-exporting countries. This proportion depends on the differences between the emission intensities of the re-exporting countries and the countries from whom they are importing. When re-exporting countries export goods from countries with higher emissions, emissions of the re-exporting countries increase in our approach that would not be the case if we do correct for re-export. Highlighting this increase in emissions is essential so that re-exporting countries would be motivated to reduce overall emissions by exporting commodities from the countries that have equal or less emissions than theirs.

Examples of such cases where re-exporting countries displayed higher emissions in our main approach compared to our third sensitivity analysis which considered re-exports are mentioned in the revised version of the results as follows:

“As an example, for paddy rice in 2015, the Netherlands and Slovenia saw deviations in TAE by almost 5 and 6 times respectively when using the ‘re-exporter’ approach. These countries re-exported around 60% of their paddy rice imports in 2015, resulting in these deviations ³³.”

- As other papers have also used the approach by Kastner et al. (2011) to correct for re-exports at comparable scales and product coverage as ours (as mentioned by the reviewer), we do not believe that repeating the same approach in our paper will provide added value to the literature on the topic. We thus argue that it would be more insightful to present our original approach as it would provide a different perspective to the debate over emission accounting instead of downplaying the significance of other approaches.

In other words, by providing an alternative approach to trade-adjusted emissions accounting, our paper complements rather than replicates or dismisses other methods of emissions accounting that have already been established in the literature. This would allow scholars and practitioners to discern how different emissions accounting approaches can produce different results for different purposes and under various accounting capacities and data availability.

We hint towards this in the following text in the revised discussion:

“By introducing our emissions accounting approach into the debate, we therefore extend the findings of previous studies that have also used ‘trade-adjustment’ approaches to estimate emissions ^{16–19,35}.”

We do however agree that it would be beneficial for us to compare our results with those of other approaches. As such, we took the additional step of contacting the leading authors of the recommended paper Romanello et al. (2021) (i.e., Dr. Ian Hamilton and Dr. Marina Romanello) to better understand the data used for the Lancet study, and to also access their original data in order to make the comparisons. Both authors referred us to Dr. Harry Kennard and Dr. Carole Dalin, who have been extremely helpful in supporting us with data provision. This support enabled us to compare our

results with those of Romanello et al. (2021) at the regional level (see Supplementary Figures 12-15). We also briefly described this comparison in our main text as follows:

“For example, when comparing our TAE results with those of Romanello and colleagues¹⁸, whose geographical and temporal scopes closely resemble ours, we find that the trends for all regions are largely similar (see Supplementary Fig. 12). While the differences are quite noticeable in relative terms, particularly in the Former Soviet Union and Oceania (see Supplementary Fig. 15), it should be noted that the differences are already evident when looking at production-based emissions (i.e., before trade adjustments; see Supplementary Fig. 13-14), suggesting that these discrepancies are not solely due to methodological differences in accounting for re-exports, but instead could simply be a case of using different emission factors and food groupings.”

Furthermore, in our discussions with the leading authors and researchers of the paper (Romanello et al. 2021), we were informed that the indicators used to develop the Lancet database explorer are currently undergoing further developments. This exemplifies how the various existing approaches to trade-adjusted emissions accounting are constantly being refined, and that there is always room for alternative and perhaps complementary methods to emissions accounting, to which our study could offer important contributions.

However, we would like to stress one important point: upon closer inspection of the reviewer's recommended paper Romanello et al. (2021), we noted that the paper was published online on 20 October 2021, *after* the submission of our last revised version in March 2021. Similarly, the other recommended paper Watts et al. (2020) was published in December 2020, *after* the submission of our first version in June 2020. As such, it would not have been possible for us to be aware of - let alone incorporate - the findings of the recommended papers in each corresponding revision. It therefore seems rather unjustified that our study's contributions (in its previous iteration) would be considered “too thin” as stated by the reviewer. Nevertheless, in view of this new paper by Romanello and colleagues that has been brought to our attention, we have now constructively incorporated and even compared their results to our approach and specifications in the current version of our manuscript.

Reviewer response: actually, the country-level analysis is also available in this published literature (cf. references above). Please adjust all references in the text accordingly – as suggested above, this is an opportunity to adjust the study in order for it to add more value to the literature on this topic.

Response: We thank the reviewer for pointing out that country-level analyses are already available in the cited literature. However, this was not immediately clear to us as Watts et al. (2020) only presented aggregated results. Nevertheless, to more accurately reflect the fact that country-level analyses are already available in the literature, we have made the necessary corrections to the first paragraph of the discussions as follows:

“By introducing our emissions accounting approach into the debate, we therefore extend the findings of previous studies that have also used ‘trade-adjustment’ approaches to estimate emissions^{16–19,35}.”

Reviewer: great. That may help support more added value to this study if different approaches are more extensively compared.

Response: We thank the reviewer very much for acknowledging the changes and clarifications we made in the previous revision of the manuscript regarding our study's trade policy implications and the issue of feed. Furthermore, and as mentioned above, we have now compared different approaches and specifications to emissions accounting. This includes, for example, a comparison of our results with those from Romanello et al. (2021) (described above), as well as our three sensitivity analyses. In our third sensitivity analysis, for instance, we make the comparison more explicit as follows:

“Similar to the second sensitivity analysis, the largest differences were found for the United Arab Emirates, with TAEs deviating by as much as 62 times from the original approach in 1997. Looking at specific food groups, we also found large differences among countries in Oceania, the Middle East, southern Africa, and parts of Europe (Supplementary Fig. 9). [...] However, we observed such differences in only a few cases, mainly for intermediary trading

countries, and over 91% of differences fell within +/-10% from the original approach during the period of 1986-2017.

Reviewer response: OK. You may simply say that feed crops are allocated to crop accounts rather than livestock accounts. This is another issue for trade that is worth mentioning (as feed crops can be traded before the livestock products may be trade themselves).

Response: Many thanks for the suggestion. We mentioned crops used as feed accordingly in our revised version.

Reviewer comments, third round review –

Reviewer #1 (Remarks to the Author):

Dear authors,

After Reading yours responses to reviewer´s comments and queries (i.e. Rebuttal letter) and once the revised version of the manuscript has been read and analysed, I can confirm that the MS has been substantially improved within the revision process.

Authors made a significant effort with this revision as shown by (e.g.) the new text included, new references and discussions maintained with other researchers with strong experience on the assessed matter and the methodological approach used.

For all the above mentioned, I would like to suggest the acceptance of the MS in its current (heavily revised) form.

Yours sincerely,

Alberto Sanz-Cobena

Reviewer #2 (Remarks to the Author):

Adjusting agricultural emissions for trade matters for climate change mitigation

(Manuscript ID: NCOMMS-20-24771C)

The authors have successfully addressed my previous comments and suggestions.

REVIEWER COMMENTS

Reviewer #1 (Remarks to the Author):

Dear authors,

After Reading yours responses to reviewer's comments and queries (i.e. Rebuttal letter) and once the revised version of the manuscript has been read and analysed, I can confirm that the MS has been substantially improved within the revision process.

Authors made a significant effort with this revision as shown by (e.g.) the new text included, new references and discussions maintained with other researchers with strong experience on the assessed matter and the methodological approach used.

For all the above mentioned, I would like to suggest the acceptance of the MS in its current (heavily revised) form.

Yours sincerely,

Alberto Sanz-Cobena

Response: We are pleased to know that the reviewer is satisfied with our revised manuscript. Thank you very much for your comments and support throughout the process of revising the manuscript.

Reviewer #2 (Remarks to the Author):

Adjusting agricultural emissions for trade matters for climate change mitigation

(Manuscript ID: NCOMMS-20-24771C)

The authors have successfully addressed my previous comments and suggestions.

Response: We are pleased to know that the reviewer is satisfied with our revised manuscript. Thank you very much for your comments and support throughout the process of revising the manuscript.